# DeSparsify: Adversarial Attack Against Token Sparsification Mechanisms

**Oryan Yehezkel**[*], **Alon Zolfi**[*], **Amit Baras, Yuval Elovici, Asaf Shabtai**
Department of Software and Information Systems Engineering
Ben-Gurion University of the Negev, Israel
{oryanyeh,zolfi,barasa}@post.bgu.ac.il, {elovici,shabtaia}@bgu.ac.il

## Abstract

Vision transformers have contributed greatly to advancements in the computer vision domain, demonstrating state-of-the-art performance in diverse tasks (*e.g.*, image classification, object detection). However, their high computational requirements grow quadratically with the number of tokens used. *Token sparsification* mechanisms have been proposed to address this issue. These mechanisms employ an input-dependent strategy, in which uninformative tokens are discarded from the computation pipeline, improving the model's efficiency. However, their dynamism and average-case assumption makes them vulnerable to a new threat vector – carefully crafted adversarial examples capable of fooling the sparsification mechanism, resulting in worst-case performance. In this paper, we present *DeSparsify*, an attack targeting the availability of vision transformers that use token sparsification mechanisms. The attack aims to exhaust the operating system's resources, while maintaining its stealthiness. Our evaluation demonstrates the attack's effectiveness on three token sparsification mechanisms and examines the attack's transferability between them and its effect on the GPU resources. To mitigate the impact of the attack, we propose various countermeasures. The source code is available online[1].

## 1 Introduction

In the last few years, vision transformers have demonstrated state-of-the-art performance in computer vision tasks, outperforming traditional convolutional neural networks (CNNs) in various tasks such as image classification, object detection, and segmentation [13]. While vision transformers have excellent representational capabilities, the computational demands of their transformer blocks make them unsuitable for deployment on edge devices. These demands mainly arise from the quadratic number of interactions (inter-/intra-calculations) between tokens [13]. Therefore, to reduce the computational requirements, various techniques have been proposed to improve their resource efficiency. *Token sparsification* (TS), in which tokens are dynamically sampled based on their significance, is a prominent technique used for this purpose. The TS approaches proposed include: ATS [5], AdaViT [20], and A-ViT [33], each of which adaptively allocates resources based on the complexity of the input image (*i.e.*, input-dependent inference) by deactivating uninformative tokens, resulting in improved throughput with a slight drop in accuracy. Despite the fact that TS has been proven to be effective in improving the resource efficiency of vision transformers, their test-time dynamism and average-case performance assumption creates a new attack surface for adversaries aiming to compromise model availability. Practical implications include various scenarios, such as: attacks on cloud-based IoT applications (*e.g.*, surveillance cameras) and attacks on real-time DNN inference for resource- and time-constrained scenarios (*e.g.*, autonomous vehicles) [10].

---

[*]Equal contribution
[1]https://github.com/oryany12/DeSparsify-Adversarial-Attack

38th Conference on Neural Information Processing Systems (NeurIPS 2024).

Given the potential impact of availability-oriented attacks, the machine learning research (ML) community has increasingly focused its attention on adversarial attacks aimed at compromising model availability. Shumailov et al. [27] were at the forefront of research in this emerging domain, introducing sponge examples, which leverage data sparsity to escalate GPU operations, resulting in increased inference times and energy consumption.

Exploiting this technique, Cinà et al. [3] deliberately poisoned models with sponge examples in the training phase to induce delays during the inference phase. The post-processing phase of deep neural networks (DNNs), particularly in the context of object detection [26] and LiDAR detection [15], has also been shown to be susceptible to availability-oriented attacks. In addition, dynamic neural networks (DyNNs) [7], which adapt their structures or parameters based on input during the inference phase, have been found to be vulnerable to adversarial attacks. For example, previous studies demonstrated that layer-skipping and early-exit mechanisms are also vulnerable to malicious inputs that aim to induce worst-case performance [8, 11, 24].

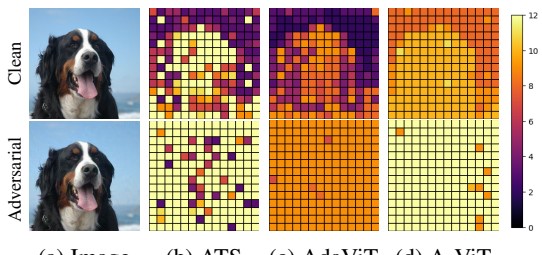

(a) Image    (b) ATS    (c) AdaViT    (d) A-ViT

Figure 1: Token depth distribution in terms of transformer blocks for a clean (top) and adversarial (bottom) image for three TS mechanisms (b)-(d). The colors indicate the maximum depth each token reaches before being discarded. The adversarial image is crafted using the single-image attack variant (Section 4.1), which results in worst-case performance.

In this paper, we introduce the *DeSparsify* attack, a novel adversarial attack that targets TS mechanisms, exploiting their test-time dynamism to compromise model availability. To perform our attack, we craft adversarial examples using a custom loss function aimed at thwarting the sparsification mechanism by generating adversarial examples that trigger worst-case performance, as shown in Figure 1. To increase the stealthiness of our adversarial examples in scenarios where anomaly detection mechanisms are employed (*e.g.*, monitoring shifts in the predicted distribution), the attack is designed to preserve the model's original classification. The experiments performed in our comprehensive evaluation examine the attack's effect for: *(i)* different sparsification mechanisms (*i.e.*, ATS, AdaViT, and A-ViT); *(ii)* different transformer models (*i.e.*, DeiT [30], T2T-ViT [34]); *(iii)* compare the performance of different attack variations (*i.e.*, single-image, class-universal, and universal); and *(iv)* investigate the adversarial examples' transferability between different TS mechanisms and the effect of ensembles. For example, the results of our attack against ATS show that it can increase the number of floating-point operations by 74%, the memory usage by 37%, and energy consumption by 72%.

Our contributions can be summarized as follows:

- To the best of our knowledge, we are the first to both identify TS mechanisms dynamism as a threat vector and propose an adversarial attack that exploits the availability of vision transformers while preserving the model's original classification.

- We conduct a comprehensive evaluation on various configurations, examining different TS mechanisms and transformer models, reusable perturbations, transferability, and the use of ensembles.

- We discuss countermeasures that can be employed to mitigate the threat posed by our attack.

## 2   Background

Vision transformers [33, 30, 34] consist of a backbone network, which is usually a transformer encoder comprised of $L$ blocks, each of which consists of a multi-head self-attention layer (MSA) and feedforward network (FFN).

We consider a vision transformer $f : \mathcal{X} \to \mathbb{R}^M$ that receives an input sample $x \in \mathcal{X}$ and outputs $M$ real-valued numbers that represent the model's confidence for each class $m \in M$. The input image $x \in \mathbb{R}^{C \times W \times H}$ is first sliced into a sequence of $N$ 2D patches, which are then mapped into patch embeddings using a linear projection. Next, a learnable class token is appended, resulting in a sequence of size $N + 1$. Finally, positional embeddings are added to the patch embeddings to provide

positional information. A single-head attention is computed as follows:

$$\text{Attn}(Q, K, V) = \text{Softmax}\left(\frac{QK^T}{\sqrt{d_k}}\right) V = AV \tag{1}$$

where $Q, K, V \in \mathbb{R}^{(N+1) \times d}$ represent the query, key, and value matrices, respectively. These matrices are derived from the output of the previous block of the transformer, denoted as $Z_l$, where $l$ indicates the $l$-th block. For the first block, *i.e.*, $Z_0$, the value corresponds to the flattened patches of the input image mentioned above.

## 3 Related Work

### 3.1 Token sparsification (TS)

In this section, we provide an overview of the three TS mechanisms we focus on in this paper.

**Adaptive Token Sampling (ATS)** [5]. ATS is a differentiable parameter-free module that adaptively downsamples input tokens. It automatically selects an optimal number of tokens in each stage (transformer block) based on the image content. The input tokens in each block are assigned significance scores by employing the attention weights of the classification token in the self-attention layer. A key advantage of ATS is its ability to be seamlessly integrated into pretrained vision transformers without the need for additional parameter tuning.

**AdaViT** [20]. AdaViT is an end-to-end framework for vision transformers that adaptively determines the use of tokens, self-attention heads, and transformer blocks based on input images. A lightweight subnetwork (*i.e.*, a decision network) is inserted into each transformer block of the backbone, which learns to make binary decisions on the use of tokens, self-attention heads, and the block's components (MSA and FFN). The decision networks are jointly optimized with the transformer backbone to reduce the computational cost while preserving classification accuracy.

**A-ViT** [33]. A-ViT is an input-dependent spatially-adaptive inference mechanism that halts the computation of different tokens at different depths, reserving computation for discriminative tokens in a dynamic manner. This halting score-based module is incorporated into an existing vision transformer block by allocating a single neuron in the MLP layer to perform this task, *i.e.*, it does not require any additional parameters or computation for the halting mechanism.

### 3.2 Availability-oriented attacks

Confidentiality, integrity, and availability, collectively known as the CIA triad, serve as a foundational model for the design of security systems [25]. In the DNN realm, a significant amount of research performed has been devoted to adversarial attacks, particularly those focused on compromising integrity [29, 6, 21, 22, 28, 32, 36, 37] and confidentiality [1, 12]. However, adversarial attacks targeting the availability of these models have only recently begun to receive attention by the ML research community.

Pioneers in the field of availability-oriented adversarial attacks, Shumailov et al. [27], introduced sponge examples, an attack designed to compromise the efficiency of vision and NLP models. The authors presented two attacks exploiting: *(i)* data sparsity - the assumption of data sparsity, which enables GPU acceleration by employing zero-skipping multiplications, and *(ii)* computation dimensions - the internal representation size of inputs and outputs (*e.g.*, in transformers, mapping words to tokens). Both attacks aim to maximize GPU operations and memory accesses, resulting in increased inference time and energy consumption. Taking advantage of the data sparsity attack vector, Cinà et al. [3] proposed sponge poisoning, which aims to compromise a model's performance by targeting it with a sponge attack during the training phase. Boucher et al. [2] introduced a notable extension of the sponge attack (computation dimension vulnerability), presenting an adversarial strategy for NLP models. This method employs invisible characters and homoglyphs to significantly manipulate the model's output, while remaining imperceptible to human detection.

Designed to enhance computational efficiency by adapting to input data during runtime, DyNNs [7] have been shown to be susceptible to adversarial attacks. For example, Haque et al. [8] targeted DNNs employing layer-skipping mechanisms, forcing malicious inputs to go through all of the layers. Hong et al. [11] fooled DNNs that employ an early-exit strategy (dynamic depth), causing malicious

inputs to consistently bypass early exits, thereby inducing worst-case performance. In a unified approach, Pan et al. [24] proposed a method for generating adversarial samples that are capable of attacking both dynamic depth and width networks.

Another line of research focused on the post-processing phase of DNNs. Shapira et al. [26] demonstrated that overloading object detection models by massively increasing the total number of candidates input into the non-maximum suppression (NMS) component can result in increased execution times. Building on this, Liu et al. [15] extended the approach to LiDAR detection models.

In this paper, we present a novel attack vector that has not been studied before, an adversarial attack that targets the availability of efficient transformer-based models that employ TS mechanisms.

## 4 Method

### 4.1 Threat model

**Adversary's Goals.** We consider an adversary whose primary goal is to generate an adversarial perturbation $\delta$ that causes TS mechanisms to use *all* available tokens, *i.e.*, no tokens are sparsified. Furthermore, as a secondary goal, to increase the stealthiness of the attack, the adversary aims to maintain the model's original classification.

**Adversary's Knowledge.** To assess the security vulnerability of TS mechanisms, we consider three scenarios: *(i)* a white-box scenario in which the attacker has full knowledge about the victim model, *(ii)* a grey-box scenario in which the attacker has partial knowledge about the set of potential models; *(iii)* a black-box scenario in which the attacker crafts a perturbation on a surrogate model and applies it on a different victim model.

**Attack Variants.** Given a dataset $\mathcal{D}$ that contains multiple pairs $(x_i, y_i)$, where $x_i$ is a sample and $y_i$ is the corresponding label, we consider three attack variants: *(i)* single-image - a different perturbation $\delta$ is crafted for each $x \in \mathcal{D}$, *(ii)* class-universal - a single perturbation $\delta$ is crafted for a target class $m \in M$, and *(iii)* universal - a single perturbation $\delta$ is crafted for all $x \in \mathcal{D}$.

### 4.2 DeSparsify attack

To achieve the goals presented above, we utilize the PGD attack [18] with a modified loss function (a commonly used approach [26, 11]). The update of the perturbation $\delta$ in iteration $t$ is formulated as:

$$\delta^{t+1} = \prod_{||\delta||_p < \epsilon} (\delta^t + \alpha \cdot \text{sgn}(\nabla_\delta \sum_{(x,y) \in \mathcal{D}'} \mathcal{L}(x, y))) \tag{2}$$

where $\alpha$ is the step size, $\prod$ is the projection operator that enforces $||\delta||_p < \epsilon$ for some norm $p$, and $\mathcal{L}$ is the loss function. The selection of $\mathcal{D}'$ depends on the attack variant: *(i)* for the single-image variant, $\mathcal{D}' = \{(x, y)\}$; *(ii)* for the class-universal variant with a target class $m \in M$, $\mathcal{D}' = \{(x, y) \in \mathcal{D} | y = m\}$; and *(iii)* for the universal variant, $\mathcal{D}' = \mathcal{D}$.

Next, we describe the proposed custom loss function, which is formulated as follows:

$$\mathcal{L} = \mathcal{L}_{\text{atk}} + \lambda \cdot \mathcal{L}_{\text{cls}} \tag{3}$$

where $\mathcal{L}_{\text{atk}}$ is the attacking component, $\mathcal{L}_{\text{cls}}$ is the classification preservation component, and $\lambda$ is a scaling term which is empirically determined using the grid search approach.

The $\mathcal{L}_{\text{cls}}$ component, set to achieve the secondary goal, is defined as follows:

$$\mathcal{L}_{\text{cls}} = \frac{1}{M} \sum_{m=1}^{M} \mathcal{L}_{\text{CE}}(f_m(x + \delta), f_m(x)) \tag{4}$$

where $f_m$ denotes the score for class $m$ and $\mathcal{L}_{\text{CE}}$ denotes the cross-entropy loss.

The $\mathcal{L}_{\text{atk}}$ component, set to achieve the main goal, will be described separately for each TS mechanism in the subsections below.

#### 4.2.1 Attack on ATS

**Preliminaries.** To prune the attention matrix $A$, *i.e.*, remove redundant tokens, ATS [5] uses the weights $A_{\{1,2\}}, ..., A_{\{1,N+1\}}$ as significance scores, where $A_1$ represents the attention weights of the classification token, and $A_{1,j}$ represents the importance of the input token $j$ for the output classification token. The significance score for a token $j$ is thus given by: $S_j = \frac{A_{1,j} \times ||V_j||}{\sum_{i=2}^{N+1} A_{1,i} \times ||V_i||}$ where $\{i, j \in 2, \ldots, N+1\}$. For multi-head attention, the score is calculated for each head, and those scores are totaled over all the heads. Since the scores are normalized, they can be interpreted as probabilities, and the cumulative distribution function (CDF) of $S$ can be calculated as $\text{CDF}_i = \sum_{j=2}^{j=i} S_j$. Given the cumulative distribution function, the token sampling function is obtained by calculating the inverse of the CDF: $\psi(r) = \text{CDF}^{-1}(r) = n$, where $r \in [0, 1]$ and $n \in [2, N+1]$ (which corresponds to a token's index). To obtain $R$ samples, a fixed sampling scheme is used by choosing: $r = \left\{ \frac{1}{2R}, \frac{3}{2R}, \ldots, \frac{2R-1}{2R} \right\}$. If a token is sampled more than once, only one instance kept. Next, given the indices of the sampled tokens, the attention matrix is refined by only selecting the rows that correspond to the sampled tokens. For example, in the case in which a token $j$ in $S$ is assigned a high significance score, it will be sampled multiple times, which will result in less unique tokens in the final set.

**Attack Methodology.** Since our goal is to prevent the method from sparsifying any tokens, we want the sampling procedure to sample as many unique tokens as possible, *i.e.*, $R' = |\{\psi(r') | r' \in r\}| \to R$. The number of unique sampled tokens $R'$ depends on the distribution they are drawn from (the CDF of $S$). In an extreme case, when the scores are not balanced ($S_j \to 1$), only the dominant token $j$ will be sampled, *i.e.*, $R' = 1$. In another extreme case, in which the scores are perfectly balanced, each token will only be sampled once, and none of the tokens will be sparsified, resulting in $R' = R$. Therefore, we want to push the vector $S$ towards a uniform distribution, which will result in a balanced scores vector.

Formally, let $\hat{S}$ be a vector representing a uniform distribution. The loss component we propose is formulated as:

$$\mathcal{L}_{\text{ATS}} = \frac{1}{L} \sum_{l}^{L} \ell_{\text{KL}}(S^l, \hat{S}) \tag{5}$$

where $\ell_{\text{KL}}$ denotes the Kullback-Leibler (KL) divergence loss and $S^l$ denotes the scores vector of the $l$-th block. The use of KL divergence loss enforces the optimization to consider all the elements in the scores vector $S^l$ as a distribution, as opposed to a simple distance metric (*e.g.*, MSE loss) that only considers them as independent values.

#### 4.2.2 Attack on AdaViT

**Preliminaries.** In AdaViT [20], a decision network is inserted into each transformer block to predict binary decisions regarding the use of patch embeddings, self-attention heads, and transformer blocks. The decision network in the $l$-th block consists of three linear layers with parameters $W_l = \{W_l^p, W_l^h, W_l^b\}$ to produce computation usage policies for patch selection, attention head selection, and transformer block selection, respectively. Formally, given the input to the $l$-th block $Z_l$, the usage policy matrices for the block are computed as $(m_l^p, m_l^h, m_l^b) = \{W_l^p, W_l^h, W_l^b\} Z_l$, where $m_l^p \in \mathbb{R}^N, m_l^h \in \mathbb{R}^H, m_l^b \in \mathbb{R}^2$. Since the decisions are binary, the action of keeping/discarding is resolved by applying Gumbel-Softmax [17] to make the process differentiable $M_l = (GS(m_l^b), GS(m_l^h), GS(m_l^p))$, where $M_l \in \{0,1\}^{(2+H+N)}$ and $GS$ is the Gumble-Softmax function. For example, the $j$-th patch embedding in the $l$-th block is kept when $M_{l,j}^p = 1$ and dropped when $M_{l,j}^p = 0$. It should also be noted that the activation of the attention heads depends on the activation of the MSA.

**Attack Methodology.** The output of the decision network in each block $M_l$ provides a binary decision about which parts will be activated. Therefore, our goal is to push all the decision values towards the "activate" decision, which will result in no sparsification. Practically, we want the Gumbel-Softmax values to be equal to one, *i.e.*, $\{M_{l,i} = 1 | \forall i\}$.

Formally, we define the loss component as follows:

$$\mathcal{L}_{\text{AdaViT}} = \frac{1}{L} \sum_l^L \left( \frac{1}{2} \sum_b^2 \ell_{\text{MSE}}(M_l^b, \hat{M}_l^b) + \frac{\mathbb{1}_{M_l^0=1}}{H} \sum_h^H \ell_{\text{MSE}}(M_l^h, \hat{M}_l^h) + \frac{1}{N} \sum_p^N \ell_{\text{MSE}}(M_l^p, \hat{M}_l^p) \right)$$

(6)

where $\ell_{\text{MSE}}$ denotes the MSE loss, $\hat{M}_l$ denotes the target value (set at one), and $M_l^0$ denotes the decision regarding the activation of the MSA in block $l$. We condition the attention heads' term with $M_l^0$, to avoid penalizing the activation of the attention heads when the MSA is deactivated ($M_l^0 = 0$). When $M_l^0 = 0$, the attention heads in that block are also deactivated.

### 4.2.3 Attack on A-ViT

**Preliminaries.** In A-ViT [33], a global halting mechanism that monitors all blocks in a joint manner is proposed; the tokens are adaptively deactivated using an input-dependent halting score. For a token $j$ in block $l$, the score $h_j^l$ is computed as follows:

$$h_j^l = H(Z_j^l) \tag{7}$$

where $H(\cdot)$ is a halting module, and $h_j^l$ is enforced to be in the range $[0, 1]$. As the inference progresses into deeper blocks, the score is simply accumulated over the previous blocks' scores. A token is deactivated when the cumulative score exceeds $1 - \tau$:

$$I_j = \arg\min_{n \leq L} \sum_{l=1}^n h_j^l \geq 1 - \tau \tag{8}$$

where $\tau$ is a small positive constant that allows halting after one block and $I_j$ denotes the layer index at which the token is discarded. Once a token is halted in block $l$, it is also deactivated for all remaining depth $l > I_j$. The halting module $H(\cdot)$ is incorporated into a single neuron in the token's embedding – specifically, the first neuron. The neuron is "spared" from the original embedding dimension, and thus no additional parameters are introduced, enabling halting score calculation as:

$$H(Z_j^l) = \sigma(\gamma * Z_{j,0}^l + \beta) \tag{9}$$

where $Z_{j,0}^l$ indicates the first dimension of token $Z_j^l$, $\sigma(\cdot)$ denotes the logistic sigmoid function, and $\gamma$ and $\beta$ are respectively learnable shifting and scaling parameters that are shared across all tokens.

**Attack Methodology.** As noted above, the decision whether to deactivate a token $j$ in block $n$ (and for all the remaining blocks) relies on the cumulative score $\sum_{l=1}^n h_j^l$. If the cumulative score exceeds the threshold $1 - \tau$, then the token is halted. Therefore, we want the push the cumulative score for *all* blocks beneath the threshold, and practically, we want to push it towards zero (the minimum value of the sigmoid function). This will result in the use of token $j$ for all blocks, *i.e.*, $I_j \to L$. Formally, we define the loss component as:

$$\mathcal{L}_{\text{A-ViT}} = \frac{1}{N} \sum_{j=1}^N \left( \frac{1}{L} \sum_{n=1}^L \mathbb{1}_{I_j<n} \left( \ell_{\text{MSE}}\left( \sum_{l=1}^n h_j^l, 0 \right) \right) \right) \tag{10}$$

where $\mathbb{1}_{I_j<n}$ is used to avoid penalizing tokens in deeper blocks that have already been halted.

## 5 Evaluation

### 5.1 Experimental setup

**Models.** We evaluate our attack on the following vision transformer models: *(i)*; data-efficient image transformer [30] (DeiT) small size (DeiT-s) and tiny size (DeiT-t) versions; *(ii)* Tokens-to-Token ViT [34] (T2T-ViT) 19-block version. All models are pretrained on ImageNet-1K, at a resolution of 224x224, where the images are presented to the model as a sequence of fixed-size patches (resolution 16x16).

**Datasets.** We use the ImageNet [4] and CIFAR-10 [14] datasets, and specifically, the images from their validation sets, which were not used to train the models described above. For the single-image

attack variant, we train and test our attack on 1,000 random images from various class categories. For the class-universal variant, we selected 10 random classes, and for each class we train the perturbation on 1,000 images and test them on unseen images from the same class. Similarly, for the universal variant, we follow the same training and testing procedure, however from different class categories.

**Metrics.** To evaluate the effectiveness of our attack, we examine the following metrics:

- **Token Utilization Ratio (TUR)**: the ratio of active tokens (those included in the computation pipeline during model inference) to the total number of tokens in the vision transformer model.

- **Memory Consumption**: the GPU memory usage during model inference.

- **Throughput**: the amount of time it takes the model to process an input and produce the output.

- **Energy Consumption**: the overall GPU power usage during inference. This metric provides insights into the attack's influence on energy efficiency and environmental considerations.

- **Giga Floating-Point Operations per Second (GFLOPS)**: the number of floating-point operations executed by the model per second.

- **Accuracy**: the performance of the model on its original task.

It should be noted that AdaViT and A-ViT only zero out the redundant tokens (*i.e.*, the matrices maintain the same shape), as opposed to ATS which removes them from the computation (*i.e.*, the matrices are reshaped). As a result, when evaluating the attack on AdaViT and A-ViT, the values of the hardware metrics (*e.g.*, memory, energy) remain almost identical to those of the clean images. The attack's effectiveness will only be reflected in the GFLOPS and TUR values. The GFLOPS are manually computed to simulate the potential benefit (an approach proposed in AdaViT).

**Baselines.** We compare the effectiveness of our attack to that of the following baselines:

- **Clean**: a clean image without a perturbation. We report the results for a clean image processed by both the sparsified (referred to as *clean*) and non-sparsified model (referred to as *clean w/o*). The results for the sparsified model represent the lower bound of our attack, while the results for the non-sparsified model represent the upper bound, *i.e.*, the best results our attack can obtain.

- **Random**: a random perturbation sampled from the uniform distribution $\mathcal{U}(-\epsilon, \epsilon)$.

- **Standard PGD** [18]: an attack using the model's original loss function (proposed in [11]).

- **Sponge Examples** [27]: an attack aimed at increasing the model's activation values.

**Implementation details.** In our attack, we focus on $\ell_{\mathrm{inf}}$ norm bounded perturbations, and set $\epsilon = \frac{16}{255}$, a value commonly used in prior studies [19, 23, 31, 35]. For the attack's step $\alpha$, we utilize a cosine annealing strategy [9] that decreases from $\frac{\epsilon}{10}$ to 0. We set the scaling term at $\lambda = 8 \cdot 10^{-4}$ (Equation 3). The results are averaged across three seeds. In the Appendix, we report results for other $\epsilon$ values and the ablation study we performed on the $\lambda$ value. For the TS mechanisms' hyperparameter configuration, we use the pretrained models provided by the authors and their settings. In addition to the provided pretrained models, we trained the remaining models for AdaViT and A-ViT using the same configurations. For ATS, the sparsification module is applied to blocks 4-12, and the number of output tokens of the ATS module is limited by the number of input tokens, *i.e.*, $R = 197$ in the case of DeiT-s. For AdaViT, the decision networks are attached to each transformer block, starting from the second block. For A-ViT, the halting mechanism starts after the first block. The experiments are conducted on a RTX 3090 GPU.

## 5.2   Results

Here, we present the results for DeiT-s on the ImageNet images. In the Appendix, we report the results for DeiT-t and T2T-ViT; the results on CIFAR-10; the cost of the attacks; and provide some examples for perturbed samples; Overall, we observe similar attack performance patterns for the different models and datasets.

**Effect of adversarial perturbations.** Table 1 presents the results for the various metrics of the different baselines and attack variants for the DeiT-s model when used in conjunction with each of the TS techniques. As can be seen, the baselines are incapable of compromising the sparsification mechanism. The random perturbation performs the same as a clean image, while the standard PGD

Table 1: Evaluation of DeiT-s when used with different TS modules on various baselines and attack variations. Clean w/o denotes the performance of the clean images for the non-sparsified model, and ensemble denotes perturbations that were trained with the three TS mechanisms. The number in parentheses is the percentage change between the clean and clean w/o performance.

| | Perturbation | ATS Accuracy | ATS GFLOPS | ATS TUR | AdaViT Accuracy | AdaViT GFLOPS | AdaViT TUR | A-ViT Accuracy | A-ViT GFLOPS | A-ViT TUR |
|---|---|---|---|---|---|---|---|---|---|---|
| Baselines | Clean | 88.5% | 3.09 (0%) | 0.54 (0%) | 83.6% | 2.25 (0%) | 0.53 (0%) | 92.8% | 3.57 (0%) | 0.72 (0%) |
| | Random | 85.7% | 3.10 (0%) | 0.55 (0%) | 76.2% | 2.26 (0%) | 0.54 (0%) | 91.4% | 3.56 (1%) | 0.71 (3%) |
| | Standard PGD | 1.1% | 3.07 (1%) | 0.54 (1%) | 0.5% | 2.49 (10%) | 0.59 (13%) | 1.1% | 3.64 (10%) | 0.73 (6%) |
| | Sponge Examples | 44.3% | 3.06 (5%) | 0.55 (10%) | 32.6% | 2.24 (-18%) | 0.54 (-17%) | 61.9% | 3.27 (-26%) | 0.66 (-17%) |
| | Clean w/o | - | 4.6 (100%) | 1.0 (100%) | - | 4.6 (100%) | 1.0 (100%) | - | 4.6 (100%) | 1.0 (100%) |
| Ours | Single | 88.2% | **4.20 (74%)** | **0.88 (75%)** | 82.5% | 3.27 (44%) | 0.75 (46%) | 91.8% | 4.6 (100%) | 1.0 (100%) |
| | Ensemble (Single) | 85.6% | 3.83 (50%) | 0.78 (52%) | 79.5% | 3.16 (38%) | 0.74 (44%) | 86.8% | 4.52 (93%) | 0.98 (94%) |
| | Class-Universal | 83.7% | 3.40 (21%) | 0.63 (22%) | 80.0% | 2.94 (30%) | 0.69 (33%) | 79.8% | 4.07 (49%) | 0.85 (59%) |
| | Universal | 84.4% | 3.31 (14%) | 0.62 (15%) | 77.6% | 2.71 (19%) | 0.64 (20%) | 85.6% | 3.85 (25%) | 0.83 (42%) |
| | Universal Patch | 4.6% | 3.68 (40%) | 0.73 (42%) | 20.0% | 3.00 (32%) | 0.70 (35%) | 71.0% | **4.6 (100%)** | **1.0 (100%)** |

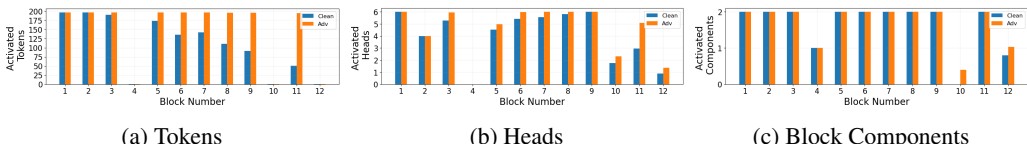

(a) Tokens      (b) Heads      (c) Block Components

Figure 2: Distribution of the (a) tokens, (b) attention heads, and (c) blocks for the AdaViT mechanism when tested on clean and adversarial (single-image variant) images.

does not affect ATS and only slightly affects AdaViT and A-ViT. The sponge examples, on the other hand, generates perturbations that perform even worse than the clean images, *i.e.*, additional tokens are sparsified. The single-image attack variant, in which a perturbation is tailored to each image, results in the greatest performance degradation, increasing the GFLOPS values by 74%, 44%, and 100% for the ATS, AdaViT, and A-ViT, respectively. Note that the crafted perturbations have just a minor effect on the model's classification accuracy.

In general, the A-ViT is the most vulnerable of the three modules, and in this case, our attack achieves near-perfect results, increasing the TUR from 72% to 100%, *i.e.*, no tokens are sparsified. The attack's success can be attributed to the fact that A-ViT utilizes a single neuron for the sparsification mechanism, easily bypassed by our attack.

While the attack's performance against AdaViT is the least dominant among the examined TS mechanisms, further analysis reveals that this stems from the overall behavior of its decision network. Figure 2a presents the distribution of the tokens used in each of the transformer blocks. As can be seen, even on clean images, the AdaViT mechanism does not use any tokens in blocks 4, 10, and 12. The same behavior is seen in the attention heads in block 4 (Figure 2b) and in the block's components in block 10 (Figure 2c). This phenomena is also evident in the performance of our attack, which is unable to increase the number of tokens used in these blocks. In the remaining blocks, our attack maximizes the use of tokens. This phenomena could be attributed to the fact that the decision networks in these blocks are overfitted or that these transformer blocks are redundant and do not affect the classification performance even when no sparsification is applied (*i.e.*, a vanilla model).

The distribution of the tokens in the different blocks of ATS is presented in Figure 3. When tested on clean images, the ATS module gradually decreases the number of tokens used as the computation progresses to deeper blocks. However, when tested on attacked images, the distribution's mean shifted towards a higher value compared to the clean case, resulting in a large number of tokens used across all blocks. Interestingly, we have seen a special trend in which clean images whose GFLOPS are on the lower end of the spectrum (*i.e.*, "easy" images that require less tokens to be correctly classified) are affected by our attack

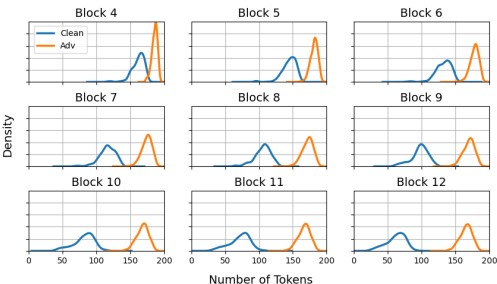

Figure 3: Distribution of activated tokens in each ATS block on clean and adversarial images.

more easily, while images whose GFLOPS are initially high are more hard to perturb. For example, for clean images that have an average of 2.7 GFLOPS, our attack is able to increase the GFLOPS to 4.2. On the other hand, clean images that have an average of 3.3 GFLOPS, the GFLOPS of the adversarial counterpart only increase to 3.9. This indicates that easily classified images tend to include more "adversarial space," in which an adversary could step in.

**Universal perturbations.** We also explore the impact of reusable perturbations – class-universal and universal perturbations (Section 4.1). In addition to the standard universal perturbation, which only differs from the single-image variant in terms of the dataset used, we explore another universal variant: a universal patch. The universal patch is trained on the same dataset used for the universal perturbation, however it differs in terms of its perturbation budget, size, and location. We place the patch on the left upper corner of the image and limit its size to $64 \times 64$, and we do not bound the perturbation budget. As seen in Table 1, the universal variants perform better than the random perturbation baseline against all sparsification modules, confirming their applicability. For example, with the class-universal perturbation, a 21%, 30%, and 49% increase in the GFLOPS values compared to the clean image was observed for the ATS, AdaViT, and A-ViT mechanisms, respectively, while maintaining relatively high accuracy. The universal patch performs even better in terms of attack performance, however its use causes a substantial degradation in the model's accuracy, resulting in a less stealthy attack. While universal and class-specific perturbations demand more resources than perturbations on single images, they possess a notable advantage. A universal perturbation has the ability to influence multiple images or an entire class of images, presenting an efficient means of executing wide-ranging adversarial attacks. This would prove to be beneficial in situations where the attacker seeks to undermine the model across numerous samples with minimal computational effort.

**Transferability and ensemble.** In the context of adversarial attacks, transferability refers to the ability of an adversarial example, generated on a surrogate model, to influence other models. In our experiments, we examine a slightly different aspect in which the effect of perturbations trained on a model with one sparsification mechanism are tested on the same model with a different sparsification mechanism. In Figure 4, we present the transferability results between the TS mechanisms. While perturbations that are trained on ATS and A-ViT, and tested on AdaViT work to some extent, other combinations do not fully transfer. We hypothesize that this occurs due to the distinct mechanism used by each model. Another strategy we evaluate is the ensemble training strategy. In this strategy, the adversarial example is trained concurrently on all of the sparsification mechanisms, and in each training iteration a different mechanism is randomly selected. The goal is to explore the synergistic advantages of leveraging the strengths of multiple models to generate adversarial perturbations that are more broadly applicable. The results of the ensemble training, which are also presented in Figure 4, show that when a perturbation is trained on all TS mechanisms, it is capable of affecting all of them, achieving nearly

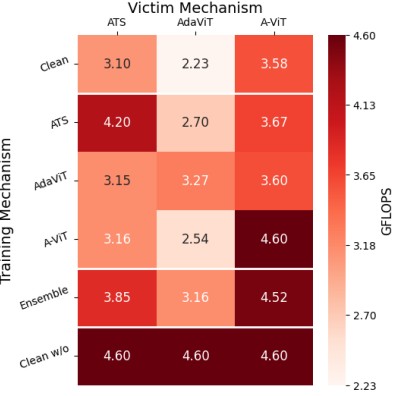

Figure 4: GFLOPS transferability results for the single-image variant attack. Ensemble refers to perturbations that were trained on all modules simultaneously.

the same performance as when the perturbations are trained and tested on the same mechanism. In a realistic case in which the adversary has no knowledge or only partial knowledge of the TS mechanism used, the ensemble is an excellent solution.

**Attack's effect on hardware.** We also assess the effect of our attack on hardware, based on several GPU metrics. As noted in Section 5.1, we only report the hardware metric results for the ATS mechanism, as it is the only module that sparsifies the tokens in practice. As seen in Table 2, which presents the results for these metrics, we can see that none of the baselines have an effect on the GPU's performance. As opposed to the baselines, the single-image attack variant increases the memory usage by 37%, the energy consumption by 72%, and the throughput by 8% compared to the clean images. As noted by the authors [5], the activation of ATS introduces a small amount of overhead associated with I/O operations performed by the sampling algorithm, which translates to longer execution time compared to the vanilla model.

Table 2: Attacks and baselines performance in terms of GPU hardware metrics for ATS. The number in parentheses represents the change from the clean images' performance.

| | Perturbation | Memory [Mbits] | Energy [mJ] | Throughput [ms] |
|---|---|---|---|---|
| **Baselines** | Clean | 240(1.00×) | 2663 (1.00×) | 12.8(1.00×) |
| | Random | 241(1.00×) | 2549(0.95×) | 12.9(1.01×) |
| | Standard PGD | 238(0.99×) | 2679(1.00×) | 12.9(1.01×) |
| | Sponge Examples | 239(1.03×) | 2865(1.07×) | 12.9(1.01×) |
| | Clean w/o | 277(1.15×) | 3020(1.13×) | 10.9(0.85×) |
| **Ours** | Single | 329(1.37×) | 4595(1.72×) | 13.8(1.08×) |
| | Ensemble (Single) | 295(1.23×) | 3587(1.34×) | 13.1(1.03×) |
| | Class-Universal | 261(1.08×) | 3926(1.47×) | 13.3(1.04×) |
| | Universal | 250(1.04×) | 3404(1.27×) | 13.1(1.03×) |
| | Universal Patch | 280(1.16×) | 4125(1.55×) | 13.4(1.05×) |

## 6    Countermeasures

In response to the challenges posed by DeSparsify, we discuss potential mitigations that can be used to enhance the security of vision transformers that utilize TS mechanisms. In general, based on our evaluation, we can conclude that as the number of parameters involved in the computation of the TS mechanism increases, the model's robustness to the attack grows (*e.g.*, a decision network in AdaViT compared to a single neuron in A-ViT). However, when the TS mechanism is based on parameters that were not optimized for this specific goal (*e.g.*, ATS attention scores), the model is even less vulnerable. To actively mitigate the threats, an upper bound can be set to the number of tokens used in each transformer block, which can be determined by computing the average number of active tokens in each block on a holdout set. This approach preserves the ability to sparsify tokens while setting an upper bound, balancing the trade-off between performance and security. To verify the validity of this approach, we evaluated two different policies for the token removal when the upper bound is surpassed: random and confidence-based policy. The results show that the proposed approach substantially decreases the adversarial capabilities compared to a clean model, *i.e.*, adversarial image GFLOPS are almost reduced the level of clean images. For example, the GFLOPS reduce from 4.2 to 3.17 when tested on ATS (clean images GFLOPS are 3.09). Moreover, we also verified that applying the defense mechanism does not degrade the accuracy on clean images. See the Appendix for details.

## 7    Conclusion

In this paper, we highlighted the potential risk vision transformers deployed in resource-constrained environments face from adversaries that aim to compromise model availability. Specifically, we showed that vision transformers that employ TS mechanisms are susceptible to availability-based attacks, and demonstrated a practical attack that targets them. We performed a comprehensive evaluation examining the attack's impact on three TS mechanisms; various attack variants and the use of ensembles were explored. We also investigated how the attack affects the system's resources. Finally, we discussed several approaches for mitigating the threat posed by the attack. In future work, we plan to explore the attack's effect in other domains (*e.g.*, NLP).

**Limitations**. A key limitation of our work is the limited transferability of the attack across different TS mechanisms and models, as it only achieves marginal success. Although we address this by proposing an ensemble training approach, future research could investigate the development of a unified loss function that effectively targets all TS mechanisms.

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

# Appendix

## A    Additional results

### A.1    Results for different models

In this section, we present the results for the DeiT-t and T2T-ViT models. Table 3 contains the results for DeiT-t and Table 4 contains the results for T2T-ViT. In Tables 5 and 6 we present the hardware performance for DeiT-t and T2T-ViT, respectively. Overall, The results show similar performance to those of the DeiT-s model, demonstrating the generalizability of our method. Note that when training DeiT-t with the AdaViT mechanism, the sparsified model is not capable of maintaining a similar level of accuracy as the non-sparsified model, using both the hyperparametes proposed by the authors and other hyperparameters experimented by us.

Table 3: Evaluation of DeiT-t when used with different sparsification mechanisms on various baselines and attack variations. Clean w/o denotes the performance of the clean images for the non-sparsified model, and ensemble denotes perturbations that were trained with the three TS mechanisms. The number in parentheses is the percentage change between the clean and clean w/o performance.

|  | Perturbation | ATS | | | AdaViT | | | A-ViT | | |
|---|---|---|---|---|---|---|---|---|---|---|
|  |  | Accuracy | GFLOPS | TUR | Accuracy | GFLOPS | TUR | Accuracy | GFLOP9PS | TUR |
| Baselines | Clean | 75.6% | 0.83(0%) | 0.54(0%) | 62.5% | 0.85(0%) | 0.84(0%) | 78.2% | 0.83(0%) | 0.65(0%) |
| | Random | 70.0% | 0.89(16%) | 0.61(15%) | 47.1% | 0.84(-2%) | 0.83(-6%) | 74.1% | 0.83(0%) | 0.65(0%) |
| | Standard PGD | 1.5% | 0.82(-2%) | 0.53(-2%) | 0% | 0.84(-2%) | 0.83(-6%) | 0% | 0.84(2%) | 0.66(2%) |
| | Sponge Examples | 38.5% | 0.82(-2%) | 0.52(-4%) | 20.4% | 0.83(-5%) | 0.81(-18%) | 31.9% | 0.80(-8%) | 0.63(-5%) |
| | Clean w/o | - | 1.2(100%) | 1.0(100%) | - | 1.2(100%) | 1.0(100%) | - | 1.2(100%) | 1.0(100%) |
| Ours | Single | 74.3% | 1.13(81%) | 0.87(72%) | 49.4% | 0.9(15%) | 0.96(75%) | 78.0% | 1.20(100%) | 0.99(97%) |
| | Class-Universal | 63.8% | 0.94(30%) | 0.65(24%) | 70.0% | 0.9(15%) | 0.93(56%) | 58.4% | 0.93(27%) | 0.73(23%) |
| | Universal | 62.9% | 0.90(19%) | 0.60(13%) | 46.5% | 0.88(9%) | 0.90(38%) | 65.1% | 0.89(17%) | 0.71(18%) |
| | Universal Patch | 1.0% | 1.04(57%) | 0.77(50%) | 5.2% | 0.9(15%) | 0.93(56%) | 71.5% | 1.20(100%) | 0.99(97%) |

Table 4: Evaluation of T2T-ViT when used with different sparsification mechanisms on various baselines and attack variations. Clean w/o denotes the performance of the clean images for the non-sparsified model, and ensemble denotes perturbations that were trained with the three TS mechanisms. The number in parentheses is the percentage change between the clean and clean w/o performance.

|  | Perturbation | ATS | | | AdaViT | | | A-ViT | | |
|---|---|---|---|---|---|---|---|---|---|---|
|  |  | Accuracy | GFLOPS | TUR | Accuracy | GFLOPS | TUR | Accuracy | GFLOP9PS | TUR |
| Baselines | Clean | 92.6% | 5.21(0%) | 0.45(0%) | 91.2% | 3.92(0%) | 0.48(0%) | 81.7% | 7.05(0%) | 0.66(0%) |
| | Random | 89.8% | 5.18(-1%) | 0.43(-3%) | 80.1% | 3.85(-1%) | 0.47(-1%) | 76.7% | 7.05(0%) | 0.65(-3%) |
| | Standard PGD | 0% | 4.99(-7%) | 0.41(-7%) | 0% | 4.15(5%) | 0.52(7%) | 0% | 7.08(2%) | 0.79(38%) |
| | Sponge Examples | 66.6% | 4.86(-10%) | 0.39(-11%) | 60.2% | 3.88(-1%) | 0.48(0%) | 0% | 6.78(-18%) | 0.70(11%) |
| | Clean w/o | - | 8.5(100%) | 1.0(100%) | - | 8.5(100%) | 1.0(100%) | - | 8.5(100%) | 1.0(100%) |
| Ours | Single | 91.6% | 7.41(67%) | 0.82(68%) | 87.6% | 5.91(44%) | 0.8(62%) | 81.7% | 8.5(100%) | 0.99(97%) |
| | Class-Universal | 85.7% | 5.90(20%) | 0.56(20%) | 82.1% | 5.38(32%) | 0.66(35%) | 67.8% | 7.26(15%) | 0.87(62%) |
| | Universal | 84.2% | 5.53(10%) | 0.51(11%) | 81.1% | 4.89(22%) | 0.61(25%) | 70.0% | 7.23(13%) | 0.81(45) |
| | Universal Patch | 87.7% | 5.77(17%) | 0.55(19%) | 73.0% | 5.12(27%) | 0.64(31%) | 70.2% | 8.4(99%) | 0.98(99%) |

Table 5: Performance of the attacks and baselines in terms of GPU hardware metrics for the ATS mechanism with DeiT-t architecture. The number in parentheses represents the percentage change from the clean images' performance.

|  | Perturbation | Memory [Mbits] | Energy [mJ] | Throughput [ms] |
|---|---|---|---|---|
| Baselines | Clean | 149(1.00×) | 2187(1.00×) | 10.3(1.00×) |
| | Random | 153(1.02×) | 3355(1.53×) | 10.5(1.03×) |
| | Standard PGD | 153(1.02×) | 3514(1.60×) | 10.3(1.00×) |
| | Sponge Examples | 146(0.97×) | 3564(1.62×) | 10.3(1.00×) |
| | Clean w/o | 163(1.10×) | 3046(1.39×) | 7.7(0.74×) |
| Ours | Single | 191(1.28×) | 3832(1.75×) | 11.3(1.10×) |
| | Class-Universal | 172(1.15×) | 3667(1.67×) | 10.7(1.05×) |
| | Universal | 155(1.18×) | 3557(1.63×) | 10.5(1.03×) |
| | Universal Patch | 176(1.18×) | 3660(1.68×) | 10.9(1.07×) |

Table 6: Performance of the attacks and baselines in terms of GPU hardware metrics for the ATS mechanism with T2T-ViT architecture. The number in parentheses represents the percentage change from the clean images' performance.

|  | Perturbation | Memory [Mbits] | Energy [mJ] | Throughput [ms] |
|---|---|---|---|---|
| Baselines | Clean | 477(1.00×) | 4562(1.00×) | 20.6(1.00×) |
| | Random | 477(1.00×) | 4671(1.02×) | 20.5(0.99×) |
| | Standard PGD | 461(0.96×) | 5085(1.11×) | 20.3(0.98×) |
| | Sponge Examples | 453(0.94×) | 5149(1.12×) | 20.1(0.97×) |
| | Clean w/o | 680(1.42×) | 6208(1.36×) | 16.6(0.80×) |
| Ours | Single | 654(1.37×) | 6822(1.50×) | 22.8(1.11×) |
| | Class-Universal | 585(1.22×) | 5616(1.26×) | 21.8(1.06×) |
| | Universal | 560(1.18×) | 5941(1.31×) | 21.6(1.05×) |
| | Universal Patch | 570(1.20×) | 5754(1.27×) | 22.0(1.07×) |

## A.2 Results for different $\epsilon$ values

In this section, we present the results obtained when different $\epsilon$ values are used. Tables 7 and 8 contain the performance metric results for $\epsilon = \frac{32}{255}$ and $\epsilon = \frac{8}{255}$, respectively. Table 9 contains the hardware metric results for these $\epsilon$ values. Note that we omitted the universal patch results from the tables, as it does not depend on a specific $\epsilon$ (the perturbation budget is unlimited).

Table 7: Evaluation of the DeiT-s model when used with different sparsification mechanisms on various baselines and attack variants when $\epsilon = \frac{32}{255}$. Clean w/o denotes the performance of the clean images on the vanilla (non-sparsified) model. Ensemble denotes perturbations that were trained with the three TS mechanisms. The number in parentheses is the normalized percentage change between the clean and clean w/o performance.

| | Perturbation | ATS | | | AdaViT | | | A-ViT | | |
| | | Accuracy | GFLOPS | TUR | Accuracy | GFLOPS | TUR | Accuracy | GFLO9PS | TUR |
|---|---|---|---|---|---|---|---|---|---|---|
| Baselines | Clean | 79.7% (100%) | 3.09 (0%) | 0.54 (0%) | 77.3% (100%) | 2.26 (0%) | 0.54 (0%) | 78.6% (100%) | 3.57 (0%) | 0.71 (0%) |
| | Standard PGD | 0.0% (0%) | 2.99 (-1%) | 0.51 (-1%) | 0.0% (0%) | 2.51 (10%) | 0.60 (13%) | 0.0% (0%) | 3.7 (12%) | 0.74 (10%) |
| | Sponge Examples | 37.4% (47%) | 2.90 (-12%) | 0.49 (-10%) | 32.4% (42%) | 2.68 (17%) | 0.63 (19%) | 19.6% (25%) | 3.28 (-28%) | 0.65 (-20%) |
| | Random | 76.5% (96%) | 3.10 (0%) | 0.55 (2%) | 74.2% (96%) | 2.26 (0%) | 0.55 (1%) | 77.0% (99%) | 3.58 (0%) | 0.72 (3%) |
| | Clean w/o | - | 4.6 (100%) | 1.0 (100%) | - | 4.6 (100%) | 1.0 (100%) | - | 4.6 (100%) | 1.0 (100%) |
| Ours | Single | 78.5% (98%) | 4.42 (89%) | 0.95 (90%) | 77.1% (99%) | 3.30 (45%) | 0.75 (46%) | 78.5% (99%) | 4.6 (100%) | 1.0 (100%) |
| | Ensemble (Single) | 67.1% (84%) | 4.05 (64%) | 0.84 (66%) | 64.8% (83%) | 3.29 (44%) | 0.75 (45%) | 75.7% (96%) | 4.59 (99%) | 0.99 (97%) |
| | Universal | 55.3% (69%) | 3.36 (18%) | 0.63 (20%) | 49.6% (64%) | 2.92 (29%) | 0.68 (31%) | 48.1% (61%) | 4.1 (53%) | 0.90 (66%) |
| | Class-Universal | 21.2% (26%) | 3.56 (32%) | 0.70 (35%) | 20.4% (26%) | 3.10 (36%) | 0.72 (40%) | 26.0% (33%) | 4.37 (78%) | 0.95 (83%) |

Table 8: Evaluation of the DeiT-s model when used with different sparsification mechanisms on various baselines and attack variations when $\epsilon = \frac{8}{255}$. Clean w/o denotes the performance of the clean images on the vanilla (non-sparsified) model. Ensemble denotes perturbations that were trained with the three TS mechanisms. The number in parentheses is the normalized percentage change between the clean and clean w/o performance.

| | Perturbation | ATS | | | AdaViT | | | A-ViT | | |
| | | Accuracy | GFLOPS | TUR | Accuracy | GFLOPS | TUR | Accuracy | GFLOPS | TUR |
|---|---|---|---|---|---|---|---|---|---|---|
| Baselines | Clean | 79.7% (100%) | 3.09 (0%) | 0.54 (0%) | 77.3% (100%) | 2.26 (0%) | 0.54 (0%) | 78.6% (100%) | 3.57 (0%) | 0.71 (0%) |
| | Standard PGD | 0.0% (0%) | 3.09 (0%) | 0.54 (0%) | 0.0% (0%) | 2.45 (8%) | 0.59 (10%) | 0.0% (0%) | 3.63 (7%) | 0.73 (6%) |
| | Sponge Examples | 73.0% (91%) | 3.02 (-4%) | 0.52 (-4%) | 76.0% (98%) | 2.58 (13%) | 0.61 (15%) | 77.1% 98%) | 3.35 (-21%) | 0.67 (-13%) |
| | Random | 77.5% (97%) | 3.09 (0%) | 0.54 (0%) | 76.2% (98%) | 2.26 (2%) | 0.55 (0%) | 78.1% (99%) | 3.58 (3%) | 0.72 (0%) |
| | Clean w/o | - | 4.6 (100%) | 1.0 (100%) | - | 4.6 (100%) | 1.0 (100%) | - | 4.6 (100%) | 1.0 (100%) |
| Ours | Single | 75.1% (94%) | 3.90 (54%) | 0.80 (57%) | 74.2% (96%) | 3.15 (39%) | 0.74 (44%) | 77.4% (98%) | 4.45 (86%) | 0.90 (66%) |
| | Ensemble (Single) | 65.0% (81%) | 3.62 (36%) | 0.71 (37%) | 59.6% (77%) | 2.96 (30%) | 0.71 (37%) | 63.6% (81%) | 4.25 (67%) | 0.86 (52%) |
| | Universal | 74.3% (93%) | 3.16 (5%) | 0.57 (7%) | 66.8% (86%) | 2.52 (12%) | 0.60 (14%) | 78.0% (96%) | 3.68 (11%) | 0.74 (11%) |
| | Class-Universal | 70.3% (88%) | 3.27 (12%) | 0.61 (16%) | 62.0% (80%) | 2.77 (22%) | 0.66 (27%) | 72.0% (91%) | 3.86 (29%) | 0.78 (25%) |

Table 9: Performance of the attacks and baselines in terms of the GPU hardware metrics for the ATS mechanism when used with different $\epsilon$ values. The number in parentheses represents the percentage change from the clean images' performance.

| | Perturbation | $\epsilon = 32/255$ | | | $\epsilon = 8/255$ | | |
| | | GPU Mem [Mbits] | Energy [mJ] | Throughput [ms] | GPU Mem [Mbits] | Energy [mJ] | Throughput [ms] |
|---|---|---|---|---|---|---|---|
| Baselines | Clean | 240(1.00×) | 2663(1.00×) | 12.24(1.00×) | 240(1.00×) | 2663(1.00×) | 12.24(1.00×) |
| | Standard PGD | 232(0.97×) | 2557(0.96×) | 12.06(0.98×) | 237(0.99×) | 2556(0.96×) | 12.11(0.99×) |
| | Sponge Examples | 227(0.94×) | 2551(0.95×) | 12.05(0.98×) | 235(0.98×) | 2529(0.95×) | 12.10(0.99×) |
| | Random | 245(1.02×) | 2710(1.01×) | 12.29(1.00×) | 242(1.01×) | 2689(1.01×) | 12.23(1.00×) |
| | Clean w/o | 277(1.15×) | 3020(1.13×) | 10.96(0.89×) | 277(1.15×) | 3020(1.13×) | 10.96(0.89×) |
| Ours | Single | 347(1.45×) | 3339(1.25×) | 13.47(1.10×) | 302(1.26×) | 2996(1.11×) | 12.86(1.05×) |
| | Ensemble (Single) | 312(1.30×) | 3009(1.11×) | 13.01(1.07×) | 276(1.15×) | 2930(1.10×) | 12.80(1.04×) |
| | Universal | 260(1.08×) | 2849(1.07×) | 12.52(1.02×) | 249(1.03×) | 2716(1.02×) | 12.37(1.01×) |
| | Class-Universal | 270(1.12×) | 3001(1.13×) | 12.55(1.03×) | 250(1.04×) | 2877(1.08×) | 12.38(1.01×) |

## A.3 Ablation study for scaling hyperparameter $\lambda$

To find the optimal value for the $\lambda$ hyperparameter (Equation 3), we performed an ablation study with various $\lambda$ values. The goal was to find the optimal point at which the attack's performance does not substantially degrade, and the model's classification accuracy is maintained. In Figure 5 we present the results of the ablation study. As can be seen, when $\lambda = 0$, the perturbation generated substantially reduces the classification accuracy ($\sim 20\%$). As the $\lambda$ value increases, the accuracy improves, with just a minimal affect on the GFLOPS value (a marginal 1% change). When $\lambda = 8 \cdot 10^{-4}$, the accuracy is nearly perfectly main-

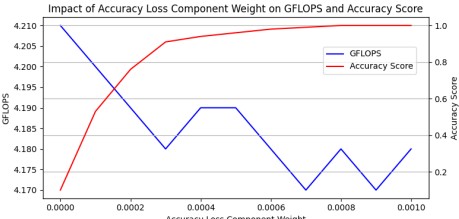

Figure 5: Ablation study examining the effect of the $\lambda$ value on the GFLOPS and accuracy for the ATS.

tained, and we consider this an optimal value. The accuracy preservation will not benefit from the use of a larger $\lambda$ value, and the use of a larger value could negatively affect the attack's performance.

## A.4 Results for other TS mechanisms

Beyond ATS, AdaViT, and A-ViT, we further demonstrate that our attack concept generalizes to additional TS mechanisms. Table 10 presents the performance of our attack on the AS-ViT mechanism [16]. Consistent with the attacks outlined in this paper, we employ a custom loss function to target AS-ViT's *Adaptive Sparsity Module*, driving the decision function to keep tokens active throughout all the transformer blocks.

Table 10: Evaluation of DeiT-s when used with AS-ViT on various baselines and attack variations. Clean w/o denotes the performance of the clean images for the non-sparsified model, and ensemble denotes perturbations that were trained with the three TS mechanisms. The number in parentheses is the percentage change between the clean and clean w/o performance.

| Perturbation | Accuracy | GFLOPS | TUR |
|---|---|---|---|
| Clean | 89.5% | 2.90(0%) | 0.65(0%) |
| Random | 80.9% | 3.03(7%) | 0.67(5%) |
| Standard PGD | 0% | 2.97(4%) | 0.64(-3%) |
| Sponge Examples | 64.4% | 3.00(6%) | 0.66(3%) |
| Clean w/o | - | 4.6(100%) | 1.0(100%) |
| Single (Ours) | 89.5% | 3.95(62%) | 0.86(60%) |

## A.5 Adversarial sample generation cost

The computational cost for the attack's generation depends on the attack variant and the token sparsification mechanism. In Table 11 we compare the cost for generating DeSparsify samples. It should be emphasized that our objective, simultaneously affecting multiple values in intermediate layers, is far more complex than the standard misclassification task, and thus, more attack iterations are required than the commonly used values. Furthermore, while universal perturbations require more resources than perturbations on single images, they possess a notable advantage: a single perturbation is applicable for all images.

Table 11: Time to create the perturbations on the DeiT-s model. Values are time in seconds.

| Mechanism | Single | Universal |
|---|---|---|
| ATS | 15 | 976 |
| Ada-ViT | 22 | 1021 |
| A-ViT | 11 | 746 |

## A.6 Results for CIFAR-10

In this section, we present the results of our attack when using CIFAR-10 images. From Table 12, we can see that our attack's performance even improves compared to the ImageNet images. This can be attributed to the greater complexity of the image distribution in ImageNet compared to CIFAR-10.

Table 12: Evaluation of CIFAR-10 images on DeiT-s when used with different sparsification mechanisms on various baselines and attack variations. Clean w/o denotes the performance of the clean images for the non-sparsified model, and ensemble denotes perturbations that were trained with the three TS mechanisms. The number in parentheses is the percentage change between the clean and clean w/o performance. Note that in this case accuracy refers to the ability to preserve the clean image prediction.

| | Perturbation | ATS | | | AdaViT | | | A-ViT | | |
|---|---|---|---|---|---|---|---|---|---|---|
| | | Accuracy | GFLOPS | TUR | Accuracy | GFLOPS | TUR | Accuracy | GFLO9PS | TUR |
| Baselines | Clean | - | 3.16(0%) | 0.57(0%) | - | 2.35(0%) | 0.55(0%) | - | 3.51(0%) | 0.70(0%) |
| | Random | 72.5% | 3.14(-1%) | 0.56(-1%) | 66.7% | 2.34(-1%) | 0.54(-1%) | 62.4% | 3.48(-2%) | 0.69(-1%) |
| | Standard PGD | 0% | 3.00(-11%) | 0.52(-11%) | 0% | 2.41(-2%) | 0.57(4%) | 0% | 3.55(3%) | 0.76(20%) |
| | Sponge Examples | 32.1% | 2.95(-14%) | 0.50(-16%) | 25.7% | 2.57(9%) | 0.60(11%) | 5.5% | 3.56(4%) | 0.72(6%) |
| | Clean w/o | - | 4.6(100%) | 1.0(100%) | - | 4.6(100%) | 1.0(100%) | - | 4.6(100%) | 1.0(100%) |
| Ours | Single | 100% | 4.41(87%) | 0.94(86%) | 89.2% | 3.31(43%) | 0.75(43%) | 99.8% | 4.6(100%) | 1.0(100%) |
| | Universal | 5.5% | 3.64(33%) | 0.71(33%) | 7.1% | 3.10(34%) | 0.72(38%) | 1.0% | 4.46(87%) | 0.97(90%) |
| | Universal Patch | 11.6% | 3.67(36%) | 0.74(40%) | 18.6% | 3.22(39%) | 0.75(45%) | 55.2% | 4.60(100%) | 0.99(96%) |

## A.7 Transferability and ensemble training across model backbones

In addition to the TS techniques transferability experiments presented in the paper, we also conducted experiments on the transferability between different backbones (DeiT-s, DeiT-t, T2T-ViT) and the

effect of ensemble strategies (trained on all three backbones). Furthermore, to provide a more generalized perspective on the capabilities of the ensemble strategy, we trained perturbations using all three backbones and three TS techniques (for a total of nine models). This approach demonstrates the ability of an attacker with partial knowledge of the environment, *i.e.*, knowing only which set of potential models and TS techniques exist (not the exact model or TS technique) to effectively carry out the attack.

Aligning with the TS mechanisms transferability and ensemble results presented in the paper, the backbone transferability and ensemble results show similar performance. For example, the average GLOPS increase when a perturbation is trained on one model backbone and tested on another are 14%, 10%, and 9% for DeiT-t, DeiT-s, and T2T-ViT, respectively. For perturbations trained with three model backbones that use the same TS mechanism, our attack achieves a 59% increase on DeiT-t, 57% increase on DeiT-s, and a 44% increase on T2T-ViT. Finally, for the perturbations trained on all nine models provide an average 38% increase on DeiT-t, 41% increase on DeiT-s, 30% increase on T2T-ViT.

## B  Countermeasures

We implemented the proposed mitigation and found it effective against the attack. We set an upper bound to the number of tokens used in each transformer block, determined by computing the average number of active tokens in each block on a holdout set. We evaluated two different policies for the token removal when the upper bound is surpassed: random and confidence-based policy. In the random policy, tokens are randomly selected to meet the threshold criteria, while in the confidence-based policy, tokens are selected based on their significance.

In Table 13 we show the accuracy results for clean images. Interestingly, when using the confidence-based policy the accuracy even improves, indicating that the token sparsification mechanisms might not be optimal since they use tokens that can "confuse" them. In Table 14, we show the GFLOPS results for adversarial images (single-image variant). Overall, we can see that the defense mechanisms substantially decrease the adversarial capabilities compared to a model that has no defense. As opposed to the accuracy results, in which the random policy demonstrated a minor performance degradation, it introduces better defense capabilities than the confidence-based policy. We hypothesize that this might occur due to the fact that informative tokens may be removed in earlier blocks (and consequently in all the remaining blocks), as opposed to the confidence-based policy which aims to maintain the highest ranking tokens throughout the entire network.

Table 13: Accuracy results for clean images on DeiT-s with and without the proposed defense.

| Module | No Defense | Defense | |
| --- | --- | --- | --- |
| | | Confidence | Random |
| ATS | 88.6% | 88.9% | 87.4% |
| Ada-ViT | 84.2% | 84.5% | 83.3% |
| A-ViT | 93.5% | 93.5% | 92.9% |

Table 14: GFLOPS results for adversarial images on DeiT-s with and without the proposed defense.

| TS Module | No Defense | Defense | |
| --- | --- | --- | --- |
| | | Confidence | Random |
| ATS | 4.2 | 3.17 | 3.04 |
| Ada-ViT | 3.27 | 2.36 | 2.16 |
| A-ViT | 4.6 | 3.95 | 3.57 |

## C  Discussion on Practical Implications

Following the practical implications discussed in Hong et al. [10], we consider two different scenarios in which our attack is applicable in:

- **Attacks on cloud-based IoT applications**: Typically, cloud-based IoT applications (*e.g.*, virtual home assistants, surveillance cameras) run their DNN inferences in the cloud. This exclusive reliance on cloud computing places the entire computational load on cloud servers, leading to heightened communication between these servers and IoT devices. Recently, there has been a trend for bringing computationally expensive models in the cloud to edge (*e.g.*, IoT)

devices. In this setup, the cloud server exclusively handles complex inputs while the edge device handles "easy" inputs. This approach leads to a reduction in computational workload in the cloud and a decrease in communication between the cloud and edge. Conversely, an adversary can manipulate simple inputs into complex ones by introducing imperceptible perturbations, effectively bypassing the routing mechanism. In this scenario, a defender could implement denial-of-service (DoS) defenses like firewalls or rate-limiting measures. In such a setup, the attacker might not successfully execute a DoS attack because the defenses regulate the communication between the server and IoT devices within a specified threshold. However, despite this, the attacker still manages to escalate the situation by: *(i)* heightening computational demands at the edge (by processing complex inputs at the edge); and *(ii)* increasing the workload of cloud servers through processing a greater number of samples.

- **Attacks on real-time DNN inference for resource- and time-constrained scenarios**: Token sparsification mechanisms can be harnessed as a viable solution to optimize real-time DNNs inference in scenarios where resources and time are limited. For example, in real-time use cases (*e.g.*, autonomous cars) where throughput is a critical factor, an adversary can potentially violate real-time guarantees. In another case, when the edge-device is battery-powered an increased energy consumption can lead to faster battery drainage.

We also discuss practical scenarios:

- **Surveillance cameras scenario:** consider a cloud-based IoT platform that uses vision transformers with a TS mechanism to process and analyze images from a network of surveillance cameras that monitor various locations and send data to a centralized cloud server for real-time analysis and anomaly detection.
  *Attack impact:* increasing computational overhead and latency could lead to delays in detecting anomalies, potentially allowing security breaches to go unnoticed for longer periods. In a high-security environment, such a delay could have severe consequences, compromising the safety and security of the monitored locations.

- **Autonomous drones scenario:** consider autonomous drones that navigate and analyze the environment using models with TS mechanisms. For example, drones that are used for delivery services, agriculture, and surveillance.
  *Attack impact:* An adversarial attack could overload the drone's computational resources (leading to rapid battery depletion and overheating) that cause navigation errors, reduced flight time, or complete system failure. These can result operational inefficiencies or accidents, especially in complex environments where precise navigation is crucial. In critical applications, such an attack could incapacitate the device, leading to mission (e.g., rescue) failure or safety hazards.

- **Wearable health monitors scenario:** consider wearable health monitors that analyze physiological data, such as heart rate, activity levels, and sleep patterns. These devices provide real-time health insights and alerts to users.
  *Attack impact:* an attack could lead to incorrect health metrics and delayed alerts. This could affect the user's health management, potentially missing critical health events that require immediate attention.

# D   Attack Visualizations

In Figure 6, we visualize the adversarial examples from the baselines and the DeSparsify different attack variants.

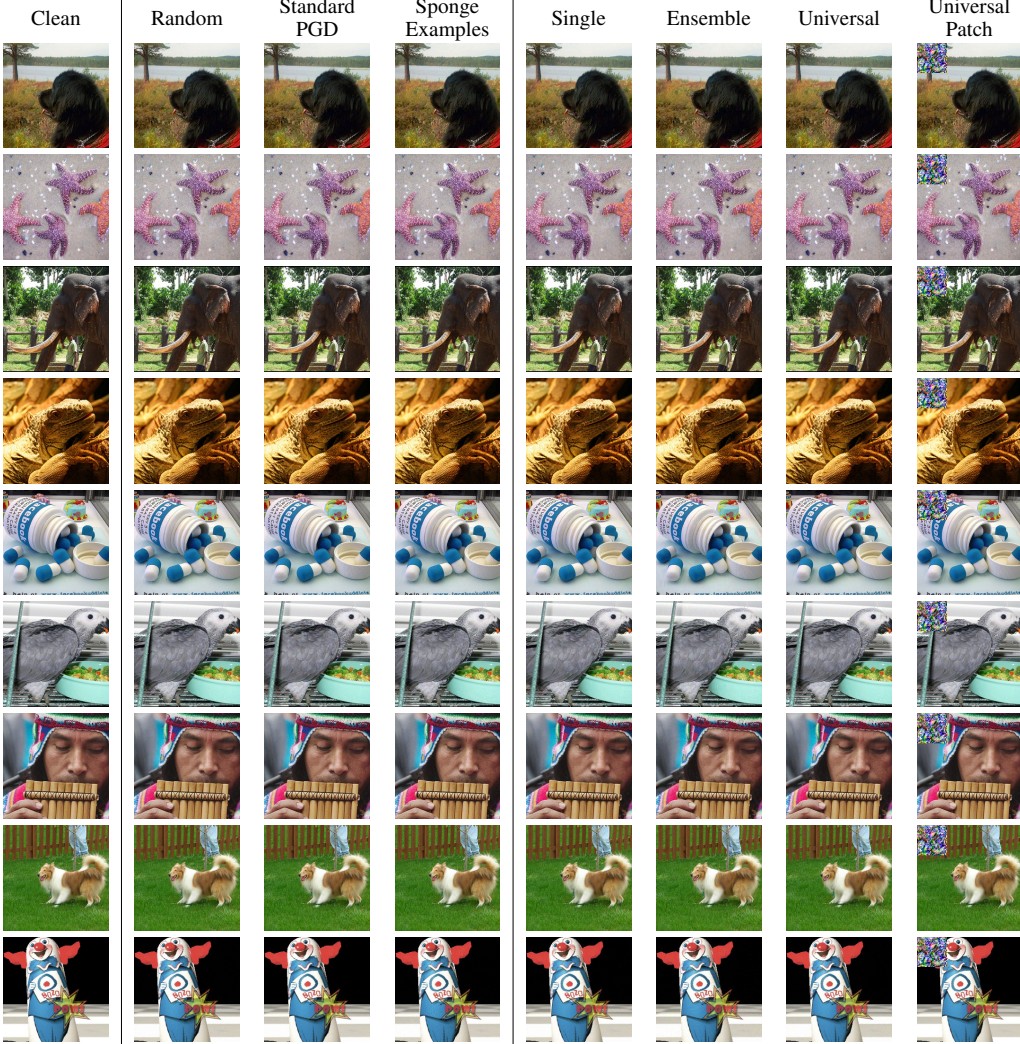

Figure 6: Adversarial examples from the baselines and our DeSrasify attacks. The leftmost column shows the clean images. In the next three columns, we show adversarial examples from random, standard PGD and sponge examples, respectively. The last four columns include adversarial examples from the different DeSparsify variants.

