# OpenReview forum: "DeSparsify: Adversarial Attack Against Token Sparsification Mechanisms"
_NeurIPS.cc/2024/Conference — NeurIPS 2024 spotlight_

### Official Review · Reviewer_vvP9 · 2024-07-03

**Soundness:** 3
**Presentation:** 3
**Contribution:** 2
**Rating:** 6
**Confidence:** 3

**Summary:**

This paper introduces DeSparsify, a novel adversarial attack targeting the availability of vision transformers that employ token sparsification techniques. The authors demonstrate that TS mechanisms, which aim to improve computational efficiency, create a new attack surface that can be exploited to compromise model availability while preserving classification accuracy. The attack is evaluated on three TS techniques (ATS, AdaViT, and A-ViT) and two vision transformer architectures (DeiT and T2T-ViT) using ImageNet and CIFAR-10 datasets. The authors explore different attack variants, including single-image, class-universal, and universal perturbations, and examine their transferability between TS techniques.

**Strengths:**

The paper presents a novel adversarial attack, addressing a previously unexplored vulnerability in token sparsification mechanisms. The authors conduct a comprehensive evaluation, testing the attack across multiple TS techniques, transformer models, and various attack variants, providing robust and detailed insights. Additionally, the paper's findings have practical implications for real-world applications, particularly in resource-constrained environments, and the proposed countermeasures add practical value, showcasing a proactive approach to addressing the identified vulnerabilities.

**Weaknesses:**

Despite its strengths, the paper has several critical weaknesses that undermine its contributions.
Firstly, the scope of the evaluation is limited to vision transformers, neglecting other domains such as natural language processing (NLP) and speech recognition. This narrow focus limits the broader impact and generalizability of the findings. The authors should have explored the applicability of DeSparsify to a wider range of models and tasks to demonstrate its broader relevance.
Secondly, the proposed countermeasures are not thoroughly compared with existing adversarial defense mechanisms. The paper presents these countermeasures in isolation, making it difficult to assess their relative effectiveness and computational overhead. A comprehensive comparison with state-of-the-art defenses would provide a clearer understanding of their strengths and weaknesses. A more in-depth review of the literature would have revealed that techniques such as SlowFormer [1] already exist.
Moreover, the transferability analysis is limited. The authors primarily focus on the transferability of the attack within different TS mechanisms in vision transformers, without exploring its applicability across diverse architectures and tasks. Understanding how the attack generalizes across different models and tasks is crucial for evaluating its robustness and broader impact.
Most importantly, the paper's contribution to the field is somewhat minimal. While identifying a new threat vector is valuable, the overall impact is reduced by the narrow focus and limited exploration of broader applicability. The attack’s success heavily relies on the assumption that the adversary has access to the model architecture and token sparsification mechanisms. While the paper evaluates both white-box and black-box scenarios, the practical feasibility of these assumptions in real-world applications are barely zero. An attacker would need to (i) find a target online vision model API, study any available documentation to understand the model's architecture, then (ii) build a surrogate model, (iii) generate adversarial examples and (iv) deploy the attack on another API (?). The methodology also appears unnecessarily complex for the problem addressed, which could have been simplified without losing effectiveness.

[1] Navaneet, K. L., Koohpayegani, S. A., Sleiman, E., & Pirsiavash, H. (2024). SlowFormer: Adversarial Attack on Compute and Energy Consumption of Efficient Vision Transformers. In Proceedings of the IEEE/CVF Conference on Computer Vision and Pattern Recognition (pp. 24786-24797).

**Questions:**

1. The success of the DeSparsify attack heavily relies on the assumption that the adversary has access to the model architecture and token sparsification mechanisms. Can you provide more details on how realistic these assumptions are in practical scenarios?
2. Can you provide more concrete examples or case studies where such an attack could be realistically deployed? Including a detailed analysis of potential real-world scenarios where this attack could be applied would add significant value. Discussing specific use cases, such as attacks on cloud-based IoT applications or real-time DNN inference, would provide clearer insights into the attack's impact and applicability.

**Limitations:**

While potential countermeasures are proposed, their practical implementation and effectiveness are not fully explored. The authors should provide more detailed experimental results or case studies demonstrating the effectiveness of the proposed countermeasures. They should also discuss any trade-offs or potential downsides of implementing these countermeasures.

The code is undocumented and references specific paths and configurations that need to exist on the file system. I was not able to reproduce results in a timely manner.

---

> ### Author Rebuttal · Authors · 2024-08-07
>
> Thank you for your time, effort and comments.
>
> **Q1**: "..limited to vision transformers.."
>
> **A1**: Our study focuses on vision transformers because the token sparsification methods we examine are specifically designed and optimized for this domain. We agree that exploring the applicability of DeSparsify to other domains such as NLP and speech recognition is important for demonstrating its broader relevance. However, extending the TS methods to these domains, first requires significant research effort for adapting them to these models, which were beyond the scope of the current study. We recognize the potential of DeSparsify in other domains and intend to pursue this direction in future work. By first establishing a solid foundation in the vision transformer domain, we aim to build a robust framework that can later be adapted and applied to a wider range of models and tasks.
>
> **Q2**: "..countermeasures are not compared.."
>
> **A2**: In our study, we have presented a comprehensive analysis of the proposed countermeasures, including two different token selection strategies, and examined their effects in terms of both availability (GFLOPS) and integrity (accuracy) on benign and adversarial samples, as detailed in Appendix B. Our results demonstrate that the proposed countermeasures successfully mitigate the impact of adversarial samples on the model's availability, reducing GFLOPS to levels close to those observed with clean images.
> Additionally, the countermeasures maintain nearly similar accuracy on clean images, indicating that they do not interfere with the model's performance on benign samples. Furthermore, to the best of our knowledge, no existing adversarial defense mechanisms specifically address availability-based attacks. Therefore, our work highlights a novel and important aspect of adversarial defenses that has not been previously explored.
>
> **Q3**: "..techniques such as SlowFormer.."
>
> **A3**: We thank the reviewer for pointing out the work on SlowFormer [1]. We were not aware of this work since it was published in conference proceedings on June 15th, while our work was submitted a month earlier on May 15th. We believe that the publication of such work in a top-tier conference emphasizes the relevance and importance of exploring the vulnerabilities of the TS domain.
> However, upon reviewing the mentioned work, we believe our research introduces novel contributions to the field, as well as offers more extensive and detailed insights. Specifically, in our study:
>
> a) we propose various attack variants, including: single, class-universal, universal, and universal patch, as opposed to SlowFormer that only evaluated a universal patch;
>
> b) we propose novel loss functions that stem from a deep understanding of each TS technique, as opposed to SlowFormer that simply uses the TS technique loss function.
>
> c) we use three different model architectures including DeiT-t, DeiT-s, and T2T-ViT.
> Some of these models were not published by the authors, which we trained ourselves.
> In SlowFormer, they only used the provided models.
>
> d) Our experiments dive deeper into each TS technique, providing novel insights and intuitions. We present evaluations on TS mechanism transferability and ensemble strategy, and show the practical effect of the attack on GPU hardware.
>
> e) we propose tailored countermeasures and present a comprehensive analysis on the trade-offs between availability and integrity.
> In addition, our countermeasure can be plugged into any TS technique without requiring model finetuning, in contrast to SlowFormer that only proposes a naive adversarial training strategy which is not clearly described.
>
> **Q4**: "..transferability analysis.." + "..assumption that the adversary has access.."
>
> **A4**: Please refer to the general comment for further details.
>
> **Q5**: "..more concrete examples.."
>
> **A5**:
>
> We discuss potential real-world scenarios where our attack on token sparsification (TS) techniques could be applied, highlighting the impact and applicability of the attack.
>
> **Surveillance cameras scenario**: consider a cloud-based IoT platform that uses vision transformers with TS techniques to process and analyze images from a network of surveillance cameras that monitor various locations and send data to a centralized cloud server for real-time analysis and anomaly detection.
>
> *Attack impact*: increasing computational overhead and latency could lead to delays in detecting anomalies, potentially allowing security breaches to go unnoticed for longer periods. In a high-security environment, such a delay could have severe consequences, compromising the safety and security of the monitored locations.
>
> **Autonomous drones scenario**: consider autonomous drones that navigate and analyze the environment using models with TS techniques. For example, drones that are used for delivery services, agriculture, and surveillance.
>
> *Attack impact*: An adversarial attack could overload the drone’s computational resources (leading to rapid battery depletion and overheating) that cause navigation errors, reduced flight time, or complete system failure. These can result operational inefficiencies or accidents, especially in complex environments where precise navigation is crucial.
> In critical applications, such an attack could incapacitate the device, leading to mission (e.g., rescue) failure or safety hazards.
>
> **Wearable health monitors scenario**: consider wearable health monitors that analyze physiological data, such as heart rate, activity levels, and sleep patterns. These devices provide real-time health insights and alerts to users.
>
> *Attack impact*: an attack could lead to incorrect health metrics and delayed alerts. This could affect the user's health management, potentially missing critical health events that require immediate attention.
>
> **Q6**: "The code.."
>
> **A6**: We have enhanced the documentation of the code, remove hard-coded paths, provided setup scripts, and created a reproducability guide.

---

> > ### Comment · Reviewer_vvP9 · 2024-08-07
> >
> > Thank you for your thoughtful and detailed rebuttal. I appreciate the effort you put into addressing my concerns, particularly the addition of practical examples and the improvements to the code documentation. These enhancements have provided valuable clarity and context to your work. Based on this, I will be adjusting my score from four to six. Thank you again for your responsiveness and the hard work you've put into this paper.

---

> > > ### Author Response · Authors · 2024-08-13
> > >
> > > Thank you for your thoughtful feedback and for acknowledging the improvements we have made. We are pleased that the addition of practical examples and the enhancements to the code documentation have provided the clarity and context you were seeking.
> > > We appreciate your careful consideration and are grateful for the opportunity to further improve our paper.

---

### Official Review · Reviewer_TQdZ · 2024-07-11

**Soundness:** 3
**Presentation:** 2
**Contribution:** 3
**Rating:** 6
**Confidence:** 4

**Summary:**

The authors investigate the scenario of an adversary forcing a vision transformer to operate less efficiently. This class of availability attack focuses on ramping up the cost of operation for the model host, who is assumed to use token sparsification (TS) to make operation cheaper. The authors formulate an attack variant of PGD which targets TS for a single sample, an entire class of samples, and the entire dataset, on different types of TS strategies, such as ATS, AdaViT, and A-ViT. The attack for ATS applies pressure through a custom loss to the significance scores distribution, so that the KL divergence of scores is closer to a uniform distribution, hence activating more tokens. The attack for AdaViT similarly applies pressure, instead forcing binary classifiers in each block towards 1 using MSE. The attack for A-ViT tries to force adaptive halting score to 0 for all block depths in the network.

**Strengths:**

- The authors investigate an interesting setting in the realm of availability attacks, which are generally under-studied. The technical details seem sound.
- The background and motivation of TS is provided in a clear way.
- The evaluation has decent breadth and incorporates logical baselines. The authors examined different attack types to reflect different attacker goals, such as attacking a single sample, an entire class, and universal perturbations. A mitigation is discussed which seems to protect the sparsification technique by setting an upper bound learned from hold-out data.
- The authors offer some novel findings. For example, experiments find that the formulated attack can increase the GFLOPS for ViTs based on the difficulty level of the image; easy images can be forced to use more power, while difficult images have lower headroom for attack. The experiments also show that universal perturbations are capable of increasing the needed computation budget of the ViT. The same was observed for single-sample and whole-class variants. An ensemble-based method manages to circumvent the three TS strategies simultaneously.
- Ablation studies are conducted in the appendices to verify some of the design decisions and choice of parameters.

**Weaknesses:**

- The submission is mainly held back by the writing quality and lack of clarity in certain sections. These are mainly focused around the formulaic description of the attacks. Presentation issues around notation and figure readability make it difficult to appreciate the otherwise interesting results.
- The main technical drawback of the attack is the need for white-box access to the model, which for the setting of cloud-hosted models, doesn't seem practical. The authors mention experiments on black-box surrogate models, but these don't seem to be in the main text nor appendices.
- The paper's threat model is mainly focused on the classification task, however ViT have seen widesprpead adoption for generative and image-to-text tasks, so there remain some open questions for these problem domains relating to vulnerability of TS.

Presentation issues:
- The choice of $K$ in section 4.2.1 (a number) clashes with $K$ (keys matrix) in Section 2. The same can be said for K'.
- The summation in the denominator of $S_j$ for section 4.2.1 should be written so it terminates at N+1, to make the index $i$ self-contained.
- In general the Section 4.2.1 preliminaries was difficult to follow, a visualization would go a long way in clarifying.
- It wasn't clear if tokens also refer to keys, e.g., if token $k$ is synonymous with key $k$. Likewise it is stated that $k \in [0, 1]$ (a real number) on L178 but then $k$ is re-defined as a set on L180, so the description is a bit confusing.
- Section 4.2.2 preliminaries: it should be clarified what $p$, $h$, and $b$ represent (patch embeddings, heads, and blocks).
- It wasn't clear what $I_j$ in Eq. 8 signifies, but it seems to imply depth due to the statement on L228.
- It would be helpful to know the standard deviation of results in Table 1.
- Figure 2 is only readable around 320% zoom, so it is too small. The authors should include axes titles to make it self-contained.

**Questions:**

1. In Section 4.2.1 it is mentioned that the token sampling function is the inverse of the CDF, but this was difficult to interpret, can the authors clarify? Mainly regarding: $CDF^{-1}(k) = n$

**Limitations:**

The authors investigate some mitigations for the proposed attack and discuss impact on certain use cases in the appendices.

---

> ### Author Rebuttal · Authors · 2024-08-07
>
> Thank you for your time, effort and comments.
>
> **Q1**: "The submission is mainly held back by the writing quality..."
>
> **A1**: Thank you for your detailed comments. We fixed the presentation issues and corrected them in the paper.
>
> **Q2**: "The main technical drawback of the attack.."
>
> **A2**: Please refer to the general comment. We have included a detailed analysis of different attacker knowledge scenarios.
>
> **Q3**: "The paper's threat model is mainly focused on the classification task.."
>
> **A3**: The focus on the classification task mainly stems from the fact that the dynamic token sparsification methods we examine in our paper have only addressed this task. Applying these methods to other domains, such as generative and image-to-text tasks, first requires significant research effort for adapting them to these models, which is beyond the scope of our current study. Nonetheless, this represents a potential area for future work, as we recognize the importance of extending our findings to the broader applications of vision transformers. It is also worth noting that since token sparsification is a novel domain, it is common practice for adversarial attacks to initially focus on image classification before being extended to other domains in subsequent research.
>
> **Q4**: "In Section 4.2.1 it is mentioned that the token sampling function..."
>
> **A4**: In other words, the significance scores are used to calculate the mapping function between the indices of the original tokens and the sampled tokens. To determine which tokens remain active in the current block, ATS uses a fixed sampling scheme $ k = \{ \frac{1}{2K}, \frac{3}{2K}, \ldots, \frac{2K-1}{2K} \} $ that is equally distributed in the range [0,1], where $ K $ is chosen to be the maximum number of tokens in block 0 (for DeiT-s, $\( K = 197 \)$, as there are 196 tokens plus an additional class token).
> For instance, if the first token has a high significance score (e.g., 0.1), the inverse of the CDF will return the index of the first token for all \( k \) values such that \( k < 0.1 \). In the example above, the first 20 entries of $k$, i.e., $\{ \frac{1}{394}, \ldots, \frac{39}{394} \}$, will map to the index of the first token. Since ATS only keeps one instance if a token is sampled more than once, the number of active tokens in the current block dramatically decreases. This will be clarified in the final version.

---

> > ### Comment · Reviewer_TQdZ · 2024-08-12
> > **Acknowledge**
> >
> > Thanks to the authors for clarifying the main issues. The new experiments offer some interesting insights on the attacker knowledge, and the presentation issues may be fixed in the next version, so I will raise my score slightly.

---

> > > ### Author Response · Authors · 2024-08-13
> > >
> > > Thank you for your feedback and for acknowledging the clarifications and new experiments. We are glad that the additional insights were helpful, and we appreciate your consideration of our work.

---

### Official Review · Reviewer_KzdF · 2024-07-12

**Soundness:** 2
**Presentation:** 3
**Contribution:** 2
**Rating:** 6
**Confidence:** 4

**Summary:**

In this paper, the authors propose an availability-oriented attack method called DeSparsify, targeting vision transformers that utilize token sparsification (TS) mechanisms.

To perform an effective attack, a custom loss function is introduced to three different ViT sparsification techniques. This approach not only exhausts the operating system's resources but also maintains the model’s original classification.

Experiments conducted on two benchmark datasets (CIFAR-10 and ImageNet) demonstrate that the proposed DeSparsify method achieves better performance across various metrics (e.g., GFLOPS, Accuracy, etc.) compared to baseline methods in different ViT architectures.

**Strengths:**

1. For purpose of attacking various TS mechanisms (i.e., Adaptive Token Samplnig (ATS), AdaViT, and A-ViT), the proposed DeSparsify method introduces a custom loss function that effectively combines the availability-oriented attack loss with the classification preservation loss.

2. This paper conducts extensive experiments to verify the effectiveness of the proposed DeSparsify method across various ViT architectures and explore the transferability of crafted adversarial examples from one sparsification mechanism to another.

3. From a defense perspective, this paper further introduces potential mitigation methods to enhance the security of ViTs based on TS mechanisms.

**Weaknesses:**

1. The proposed DeSparsify attack method utilizes projected gradient descent (PGD), as outlined in Eq. (2), to maximize the custom loss function defined in Eq. (3). However, to preserve the model’s original classification, the cross-entropy loss defined in Eq. (4) should be minimized. This objective conflicts with the operations described in Eqs. (2-3).

2. In Eqs. (5-7), this paper uses KL and MSE losses to define the availability-oriented attack loss for each ViT layer with equal weight. However, the motivation for this choice is not well explained. For example, other weighting methods, such as unequal weights, have not been discussed.

3. As described in lines 366-367, the ensemble strategy is proposed to affect all the token sparsification mechanisms. However, as shown in Tables 1-2, the ensemble strategy performs worse than the single strategy. The reason for this phenomenon is not well explained.

4. In the experiment section, this paper only evaluates the transferability of the proposed DeSparsify attack within the same ViT architecture, from one token sparsification mechanism to another, without considering the transferability between different ViT architectures.

**Questions:**

1. How do the two different components of the custom loss defined in Eq. (3) behave during the training process?
2. Can different weights for each layer, as defined in Eqs. (5-7), further enhance attack performance?
3. Can other training strategies, instead of random sampling, improve the performance of the proposed ensemble attack method?
4. Can the proposed DeSparsify attack achieve good performance across different ViT architectures?

**Limitations:**

I suggest the author include a brief discussion of the limitations of the proposed work in a separate section of this paper.

---

> ### Author Rebuttal · Authors · 2024-08-06
>
> Thank you for your time, effort and comments.
>
> **Q1**: ".. This objective conflicts with the operations described in Eqs. (2-3)."
>
> **A1**: Please note that the novel loss attacking components we propose should be minimized and not maximized.  That is, our optimization process follows the computed gradient direction, as opposed to the classic PGD attack which goes to the opposite direction. Indeed, in the implementation of the PGD baseline (denoted in the paper as Standard PGD), we negate the cross-entropy loss term to perform the attack. Therefore, there is no conflict between the operations, which is also evident in the ablation study results presented in Appendix A.3.
>
> **Q2**: "..different weights for each layer.."
>
> **A2**: The use of equal weights across the different transformer blocks is motivated by simplicity, aiming to make the design and implementation more straightforward and less complex.  This approach also ensures a fair comparison between our proposed attack and other baselines. Optimizing the weights, for instance through a grid search methodology, could enhance the effectiveness of our attack while leaving the baselines un-optimized for this specific task. Nonetheless, using unequal weights would likely improve our attack’s success when aiming for the best performance on its own, rather than in comparison to other approaches. It is also worth noting that the current attack configuration is highly effective, maximizing almost the use of all available tokens.
>
> **Q3**: "..ensemble strategy performs worse than the single strategy.."
>
> **A3**: In the ensemble strategy, the adversarial example is trained concurrently across all sparsification techniques, with a different technique randomly selected in each training iteration. In contrast, the single strategy involves training the adversarial example on a single sparsification technique. Intuitively, an adversarial example trained and tested on a single TS technique is likely to achieve the best performance, as it is specifically tailored to that technique. In comparison, the ensemble strategy must adapt the adversarial example to effectively attack multiple techniques simultaneously. Consequently, the ensemble strategy demonstrates a notable ability to affect all sparsification mechanisms to a reasonable extent.
>
> **Q4**: "..considering the transferability between different ViT architectures.."
>
> **A4**: Please refer to the general comment.
>
> **Q5**: "How do the two different components of the custom loss defined in Eq. (3) behave during the training process?"
>
> **A5**: Both terms decrease as the attack progresses through subsequent iterations, effectively achieving the dual objectives of the attack goal and preserving accuracy.
>
> **Q6**: ".. random sampling, improve the performance of the proposed ensemble attack.."
>
> **A6**: Other training strategies could be employed to further improve the performance of the ensemble attack. For example, selecting the worst-performing model at each iteration could enhance the attack's effectiveness on that particular model. However, our goal was to focus on the core concept of the ensemble approach itself, without incorporating additional complexities. We aimed for a straightforward strategy to demonstrate the fundamental effectiveness of the ensemble method.
>
> **Q7**: "..performance across different ViT architectures.."
>
> **A7**: In Appendix A.1, we show the performance of our attack on a different size DeiT (tiny) and on the T2T-ViT model. Overall, we observe similar attack performance patterns for the different ViT architectures.
>
> **Q8**: "..include a brief discussion of the limitations.."
>
> **A8**: Thank you, we will include a discussion on limitations in the final version.

---

> > ### Comment · Reviewer_KzdF · 2024-08-12
> >
> > Thanks for the authors’ responses. My concerns have been properly addressed, and I would like to raise my rating to 6.

---

> > ### Author Response · Authors · 2024-08-13
> >
> > Thank you for your follow-up and for your positive feedback. We are glad that our responses have addressed your concerns. We appreciate your time and consideration in reviewing our work.

---

### Official Review · Reviewer_yY7m · 2024-07-12

**Soundness:** 4
**Presentation:** 3
**Contribution:** 3
**Rating:** 7
**Confidence:** 4

**Summary:**

The paper presents DeSparsify, an adversarial attack on vision transformers utilizing token sparsification (TS). It highlights the vulnerability of TS techniques due to their dynamic nature and shows how DeSparsify can deplete system resources while preserving the model's original classification accuracy. The study offers a thorough evaluation of various TS techniques and transformer models and suggests countermeasures to counteract the attack.

**Strengths:**

1. This paper focus on an important research problem: the vulnerability of token sparsification technique for VLMs. The proposed attack and defense methods are inspirational for future research in related fields.
2. The proposed method is intuitive and developed with rigorous mathematical derivation. The description of the method is clear and easy to understand.
3. The proposed method achieves excellent performance in experiments and beats the state-of-the-art baseline methods.
4. The paper reports the detailed computation cost of the method, which makes it easy to be followed and adapted.

**Weaknesses:**

1. The authors can add some examples and visualizations for the adversarial examples.
2. If there were more discussion and derivation on why TS methods are vulnerable to attacks, it would make this paper more impactful.

**Questions:**

Will the code and data be publicly accessed?

**Limitations:**

The authors have discussed the limitations.

---

> ### Author Rebuttal · Authors · 2024-08-06
>
> Thank you for your time, effort and comments.
>
> **Q1**: "..add some examples and visualizations for the adversarial examples"
>
> **A1**: visualizations for the adversarial examples, including baselines and our attack variants, can be found in Appendix D. We will also include the perturbations in the final version.
>
> **Q2**: "..more discussion and derivation on why TS methods are vulnerable to attacks.."
>
> **A2**: In Section 5.2 "Effect of adversarial perturbations" we discussed several aspects that might explain the success of the attack on each one of the TS mechanisms. Furthermore, in Section 6 "Countermeasures" (lines 390-396) we provide some intuitions and comparison between the different TS mechanisms, highlighting the weak spots of each one. Overall, the main vulnerability that all the TS mechanisms posses is their test-time dynamism and average-case performance assumption, which allows potential attackers to fit in and induce worst-case performance.
>
> **Q3**: "Will the code and data be publicly accessed?"
>
> **A3**: Of course, upon acceptance the code and data will be made publicly available.

---

### Official Review · Reviewer_Ux8f · 2024-07-13

**Soundness:** 4
**Presentation:** 4
**Contribution:** 3
**Rating:** 7
**Confidence:** 4

**Summary:**

The paper proposes DeSparsify, an adversarial attack against token sparsification methods for ViTs. Such attacks aim at modifying input images to increase the inference time and cost while preserving the original classification. In particular, the paper designs specific losses against three existing sparsification mechanisms, and shows that these attacks can effectively increase the inference cost. Finally, the paper discusses potential countermeasures to defend against DeSparsify.

**Strengths:**

- The paper explores a new direction of adversarial attacks which target the sparsification mechanisms applied to vision transformers. This is one approach in the popular topic of reducing the inference cost of modern models, and studying this type of availability attacks is relevant.

- The proposed methods and losses are well-justified and presented. Moreover, the paper considers several threat models (image-specific and universal attacks).

- The experimental results show the effectiveness of the attacks in increasing the inference cost, and provides some analysis of the different performance against different methods.

**Weaknesses:**

- While I think it's important to point out these vulnerabilities, the fact the sparsification methods are not robust to adversarial attacks is not surprising.

- The attacks are only partially successful on AdaViT (44% recovery of GFLOPs without sparsification). Moreover, including more (recent) techniques (e.g. [A]) might further strengthen the paper.

[A] https://arxiv.org/abs/2209.13802

**Questions:**

- Are the (class) universal attacks tested on images different from those used to generate the perturbations? This is not clear from the text.

**Limitations:**

Yes.

---

> ### Author Rebuttal · Authors · 2024-08-06
>
> Thank you for your time, effort and comments.
>
> **Q1**: "sparsification methods are not robust to adversarial attacks is not surprising"
>
> **A1**: While it may seem unsurprising now that sparsification methods are not robust to adversarial attacks, this understanding was not trivial prior to our work. Our study provides concrete evidence and detailed analysis, which were previously lacking, thus highlighting the critical vulnerabilities in these methods. By systematically demonstrating these weaknesses, we have paved the way for future research to develop more robust sparsification techniques.
>
> **Q2**: "The attacks are only partially successful on AdaViT.."
>
> **A2**: While the success of our attack on AdaViT may seem partial, this is not actually the case. In lines 309-318 in the paper, we discuss this specific phenomena. To summarize, we show that even on a clean image, the distribution of used tokens, layers and blocks across the different transformer blocks are distorted when AdaViT is used. For example, no tokens are used in blocks 4, 10 and 12 regardless of the input sample. In these cases, our attack cannot increase the number of used tokens in these blocks as well. On the remaining blocks, our attack maximizes the number of used tokens, layers and blocks to almost 100%, showcasing that AdaViT is fully vulnerable to such attacks.
>
> **Q3**: "including more (recent) techniques (e.g. [A]) might further strengthen the paper"
>
> **A3**: We thank you for mentioning this work. We will include our attack results on this mechanism in the supplementary material final version. For the DeiT-s model, the results for the single-image variant are:
>
> | | Accuracy | GFLOPS | TUR |
> | :- | :- | :- | :- |
> | Clean |  89% | 2.98 | 0.65 |
> | Single | 89% | 3.81 | 0.83 |
>
> As can be seen from the results, our attack successfully increases the number of GFLOPS from 2.98 (on clean images) to 3.81 (on adversarial images) while maintaining the same level of accuracy.
>
> **Q4**: "..Are the (class) universal attacks tested on images different..":
>
> **A4**: The universal attacks tested images are different than those it was trained on, to showcase the perturbation's ability to transfer to unseen images. We will clarify this in the final version.

---

> > ### Comment · Reviewer_Ux8f · 2024-08-11
> >
> > I thank the authors for the response and additional experiments.
> >
> > *"While the success of our attack on AdaViT may seem partial, this is not actually the case..."*
> >
> > I would argue that this effectively means that AdaViT is robust to the attack, as the inference time is equally reduced for both clean and adversarial inputs, i.e. the attack fails.
> >
> > Overall, the paper explores an interesting direction, and presents effective attacks.

---

> > > ### Author Response · Authors · 2024-08-13
> > >
> > > Thank you for your feedback and for acknowledging the additional experiments we conducted. We appreciate your thoughtful evaluation of our work and your recognition of the direction we are exploring.

---

### Official Review · Reviewer_ZMXm · 2024-07-15

**Soundness:** 3
**Presentation:** 3
**Contribution:** 3
**Rating:** 6
**Confidence:** 2

**Summary:**

Token sparsification uses input-dependent strategy to discarded uninformative tokens, improving the resource efficiency of vision transformers. This paper propose DeSparsify, to attack vision transformers that use token sparsification. The attack aims at exhausting the operating system’s resources.

**Strengths:**

1, This paper aims at an practical and less focused topic, the new attack surface for adversaries caused by token sparsification.

2, The experiments include explorations of difference TS methods, vit backbones, and transferbility.

3, The writing is straightforward and clear, and the relevant works in introduction and related works part are also clear.

**Weaknesses:**

1, The attack goal is to causes TS techniques to use all available tokens. Compared to methods that do not use TS, is the computational cost still higher? or will the performance decrease? if not, then the attack's upperbound is a bit limited.

**Questions:**

1, can the three attack methods on ATS, AdaViT and AViT be unified?

2, In the paper, the method mentions both white box and black box in 4.1. However, since this paper tests the transferability of adversarial trained on the same model with different sparsification mechanism, this can not be regarded as black-box attack. I suggest the author claims that they mainly focused on white box attack. And it's nice if the authors can report some results on transferability of the same sparsification mechanism on different backbones.

3, In Fig.3, what's the density if no TS method is used? is the density 1?

4, compared to sponge examples [26], their energy consumption can be about 3 times the original ones, can the paper provide some explanations why the sponge exmaples and the proposed method can only be less than 2 times the original on this task?

**Limitations:**

Please refer to questions.

---

> ### Author Rebuttal · Authors · 2024-08-06
>
> Thank you for your time, effort and comments.
>
> **Q1**: "..Compared to methods that do not use TS.."
>
> **A1**: In theory, the attack's upper bound corresponds to the model's performance when no sparsification is applied, i.e., a "vanilla" model that utilizes all tokens during inference.
> In practice, in addition to the computational cost of using all tokens, the TS mechanism itself introduces additional overhead, caused by the mechanism's operations, which also influences the performance.
> While there is an upper bound, our work aims to highlight the potential risk of deploying a transformer model with a token sparsification mechanism. We seek to raise the awareness among users to an attack that can compromise the optimization of the sparsification mechanism.
>
> **Q2**: "can the three attack methods on ATS, AdaViT and AViT be unified?"
>
> **A2**: In Section 5 "Transferability and ensemble", we have presented a joint attack in the form of an ensemble, which successfully affected all token sparsification mechanisms simultaneously (see results in Figure 4 penultimate row).
> In terms of a single loss function that could affect all mechanisms, we could not find any similar characteristics that can be unified and attacked.
>
> **Q3**: "..report some results on transferability of the same sparsification mechanism on different backbones"
>
> **A3**: We thank you for your suggestion. This will be clarified in the paper's final version.
> In addition, we have conducted additional experiments that study the effect of one sparsification mechanism on different backbones.
> Please refer to the general comment for more details.
>
> **Q4**: "In Fig.3, what's the density if no TS method is used? is the density 1?"
>
> **A4**: The results in Figure 3 are evaluated on the DeiT-s model (will be clarified in the figure's caption in the final version). DeiT-s splits the image to $14 \cdot 14 = 196$ patches (tokens), with an additional class token, for a total of 197 tokens. In the case of a vanilla model (no TS method is used), there will just be a vertical line on the 197 value on the x-axis (number of tokens).
>
> **Q5**: "compared to sponge examples [26].."
>
> **A5**: In sponge examples, the authors propose attacks for the computer vision and NLP domains. While the results in the NLP domain show excellent performance, this is not the case in the computer vision domain. As reported by the authors in Section 6.2, they were only able to achieve a marginal 1-3% increase in energy consumption. Furthermore, by observing the standard deviation results in Table 3, even this increase in energy consumption does not show a clear trend. Although the paper's authors do not discuss this difference, we hypothesize that it stems from the fact that the vast majority of the computer vision models' activation values are not zero by default, as supported in Phantom Sponges [25].

---

### Author Rebuttal · Authors · 2024-08-07

First, we thank the reviewers* for their time, effort and comments.

We are pleased to see that the reviewers find that the proposed research problem is important (R1, R2, R3, R4, R5, R6), and the results and analysis are thorough (R1, R2, R3, R4, R5, R6).
We are equally glad that the reviewers have assigned a high score to the paper's soundness of the technical claims, experimental and research methodology (R1, R2, R3, R5, R6), the quality of the presentation (R1, R2, R3, R4, R6), and the quality of the overall contribution (R1, R2, R3, R5).
We have addressed the reviewers’ comments and concerns in individual responses to each reviewer. The reviews allowed us to improve our draft and the changes made in the revised draft are summarized below:
- [R1, R2, R3, R4, R5, R6]: added backbone transferability and ensemble experiments.
- [R2]: added results on another TS technique (AS-ViT).
- [R2]: clarified the use of unseen images for the universal perturbation experiments.
- [R3]: added perturbations visualizations.
- [R4]: included a brief discussion on limitations.
- [R5]: improved and corrected presentation issues.
- [R6]: discussed potential real-world scenarios.

*For brevity, we refer to reviewers ZMXm as R1, Ux8f as R2, yY7m as R3, KzdF as R4, TQdZ as R5, and vvP9 as R6, respectively.

### Attacker Knowledge Scenarios Discussion
Since our primary research goal is to explore the capabilities and limitations of our attack, we believe that the suggestions made by the reviewers are important.
Therefore, in addition to the TS techniques transferability experiments presented in the paper (Section 5.2), we also conducted experiments on the transferability between different backbones and the effect of ensemble strategies (trained on all three backbones).
Furthermore, to provide a more generalized perspective on the capabilities of the ensemble strategy, we trained perturbations using all three backbones and three TS techniques (for a total of nine models).
This approach demonstrates the ability of an attacker with partial knowledge of the environment, i.e., knowing only which set of potential models and TS techniques exist (not the exact model or TS technique) to effectively carry out the attack.

Aligning with the TS techniques transferability and ensemble results presented in the paper, the backbone transferability and ensemble results show similar performance.
For example, the average GLOPS increase when a perturbation is trained on one model backbone and tested on another are 14%, 10%, and 9% for DeiT-t, DeiT-s, and T2T-ViT, respectively.
For perturbations trained with three model backbones that use one TS technique, our attack achieves a 59% increase on DeiT-t, 57% increase on DeiT-s, and a 44% increase on T2T-ViT.
Finally, for the perturbations trained on all nine models provide an average 38% increase on DeiT-t, 41% increase on DeiT-s, 30% increase on T2T-ViT.

To sum up, the ensemble perturbation training strategy offers several distinct advantages:
1. Improved Transferability: training on an ensemble of models enhances the transferability of the generated adversarial examples, making them more likely to be effective across different architectures and settings. This is particularly valuable in black-box scenarios where the exact model details are unknown.

2. Increased Robustness: ensemble training reduces the risk of overfitting to a specific model's characteristics, resulting in a more robust attack that can generalize better across various models.

3. Scalability and Practicality: although our initial experiments assume white-box access, the use of surrogate models in an ensemble can simulate a variety of potential target models. This approach can be scaled to include more models, enhancing the attack's generalizability and making it more practical for real-world applications.

These advantages demonstrate the efficacy of the ensemble strategy in executing successful attacks even with partial knowledge of the target environment. This underscores the potential risks of deploying transformer models with token sparsification mechanisms.
We will include these results and insights in the final version.

---

### Decision · Program_Chairs · 2024-09-25

**Decision:**

Accept (spotlight)

**Comment:**

The paper tackles the problem of "adversarial attacks" against machine learning (ML) models. Specifically, the paper proposes a novel attack, named "Desparsify", which enables adversaries to target vision transformers (ViT) by affecting the computational runtime of these methods. Indeed, the attacker, instead of aiming at inducing "misclassifications" of the targeted model, seeks to increase the "cost" of the inference phase of ViT. This is an intriguing angle that has received only limited attention by prior literature on adversarial ML, which typically seek to decrease the accuracy of the attacked model. The paper is well presented and the experiments confirm the efficacy of the attack.

The reviewers agreed that the paper tackles an important research problem in a thorough way. The reviewers also found the paper to provide compelling technical content and a laudable methodology, all of which is presented in a clear and comprehensive way. The discussion phase further enabled to derive further insights that improved the already strong contribution.

For these reasons (which combine novelty from a security standpoint with technical correctness from a ML standpoint), I am recommending to accept this paper as a "spotlight".